# CRISPRi chemical genetics and comparative genomics identify genes mediating drug potency in *Mycobacterium tuberculosis*

Shuqi Li[1,5], Nicholas C. Poulton[1,5], Jesseon S. Chang[1], Zachary A. Azadian[1], Michael A. DeJesus[1], Nadine Ruecker[2], Matthew D. Zimmerman[3], Kathryn A. Eckartt[1], Barbara Bosch[1], Curtis A. Engelhart[2], Daniel F. Sullivan[2], Martin Gengenbacher[3,4], Véronique A. Dartois[3,4], Dirk Schnappinger[2] and Jeremy M. Rock[1✉]

*Mycobacterium tuberculosis* (Mtb) infection is notoriously difficult to treat. Treatment efficacy is limited by Mtb's intrinsic drug resistance, as well as its ability to evolve acquired resistance to all antituberculars in clinical use. A deeper understanding of the bacterial pathways that influence drug efficacy could facilitate the development of more effective therapies, identify new mechanisms of acquired resistance, and reveal overlooked therapeutic opportunities. Here we developed a CRISPR interference chemical-genetics platform to titrate the expression of Mtb genes and quantify bacterial fitness in the presence of different drugs. We discovered diverse mechanisms of intrinsic drug resistance, unveiling hundreds of potential targets for synergistic drug combinations. Combining chemical genetics with comparative genomics of Mtb clinical isolates, we further identified several previously unknown mechanisms of acquired drug resistance, one of which is associated with a multidrug-resistant tuberculosis outbreak in South America. Lastly, we found that the intrinsic resistance factor *whiB7* was inactivated in an entire Mtb sublineage endemic to Southeast Asia, presenting an opportunity to potentially repurpose the macrolide antibiotic clarithromycin to treat tuberculosis. This chemical-genetic map provides a rich resource to understand drug efficacy in Mtb and guide future tuberculosis drug development and treatment.

nfections caused by the bacterial pathogen *Mycobacterium tuberculosis* (Mtb) are notoriously difficult to treat. While the reasons necessitating prolonged chemotherapy are multifactorial[1,2], the intrinsic resistance of Mtb and its ability to evolve acquired resistance to all antituberculars in clinical use are major barriers to successful treatment[3,4].

While comparatively underexplored, intrinsic resistance in Mtb is typically ascribed to the low permeability of the cell envelope and the numerous efflux pumps encoded in the Mtb genome[5–7]. Acquired drug resistance in Mtb occurs via mutation and many resistance mutations have been characterized in recent decades[8]. These mutations most commonly occur in the drug target or drug activator[8–10]. Yet, our knowledge of acquired drug resistance in Mtb remains incomplete, particularly for mutations outside of the drug target or activator and which typically confer low-to-intermediate, but clinically relevant, levels of drug resistance[4,11,12].

To provide a genome-wide overview of the bacterial pathways that influence drug potency, we developed a CRISPR (Clustered Regularly Interspaced Short Palindromic Repeats) interference (CRISPRi)[13–17] chemical-genetics platform to titrate the expression of nearly all Mtb genes (essential and non-essential) and quantify bacterial fitness in the presence of different drugs. This approach identified hundreds of genes whose inhibition altered fitness during drug treatment. Mining this dataset, we discovered diverse mechanisms of intrinsic drug resistance that can be targeted to potentiate therapy. Overlaying the chemical-genetic results with

comparative genomics of Mtb clinical isolates, we identified previously unknown, clinically relevant mechanisms of acquired drug resistance. Lastly, we make the unexpected discovery of 'acquired drug sensitivities', whereby loss-of-function mutations in intrinsic drug resistance genes render some Mtb clinical strains hypersusceptible to clarithromycin, an antibiotic not typically used to treat tuberculosis (TB).

## Results

**Defining genetic determinants of drug potency with CRISPRi.** To define genes that influence drug potency in Mtb, we performed 90 CRISPRi screens across nine drugs in H37Rv Mtb. These screens used a genome-scale CRISPRi library[15] to enable titratable knockdown for nearly all Mtb genes, including protein coding genes and non-coding RNAs (Fig. 1a). Knockdown tuning enabled hypomorphic silencing of in vitro essential genes[15], thereby allowing quantification of chemical-genetic interactions for both essential and non-essential genes to provide a global overview of gene–drug interactions in Mtb.

Drugs were chosen to represent clinically relevant antituberculars as well as two drugs not traditionally used to treat TB (Supplementary Table 1). Drugs were screened at concentrations spanning the predicted minimum inhibitory concentration (MIC). After library outgrowth, we collected genomic DNA from cultures treated with three descending doses of partially inhibitory drug concentrations (Extended Data Fig. 1) and analysed sgRNA

[1]Laboratory of Host-Pathogen Biology, The Rockefeller University, New York, NY, USA. [2]Department of Microbiology and Immunology, Weill Cornell Medicine, New York, NY, USA. [3]Center for Discovery and Innovation, Hackensack Meridian Health, Nutley, NJ, USA. [4]Hackensack Meridian School of Medicine, Hackensack Meridian Health, Nutley, NJ, USA. [5]These authors contributed equally: Shuqi Li, Nicholas C. Poulton. ✉e-mail: rock@rockefeller.edu

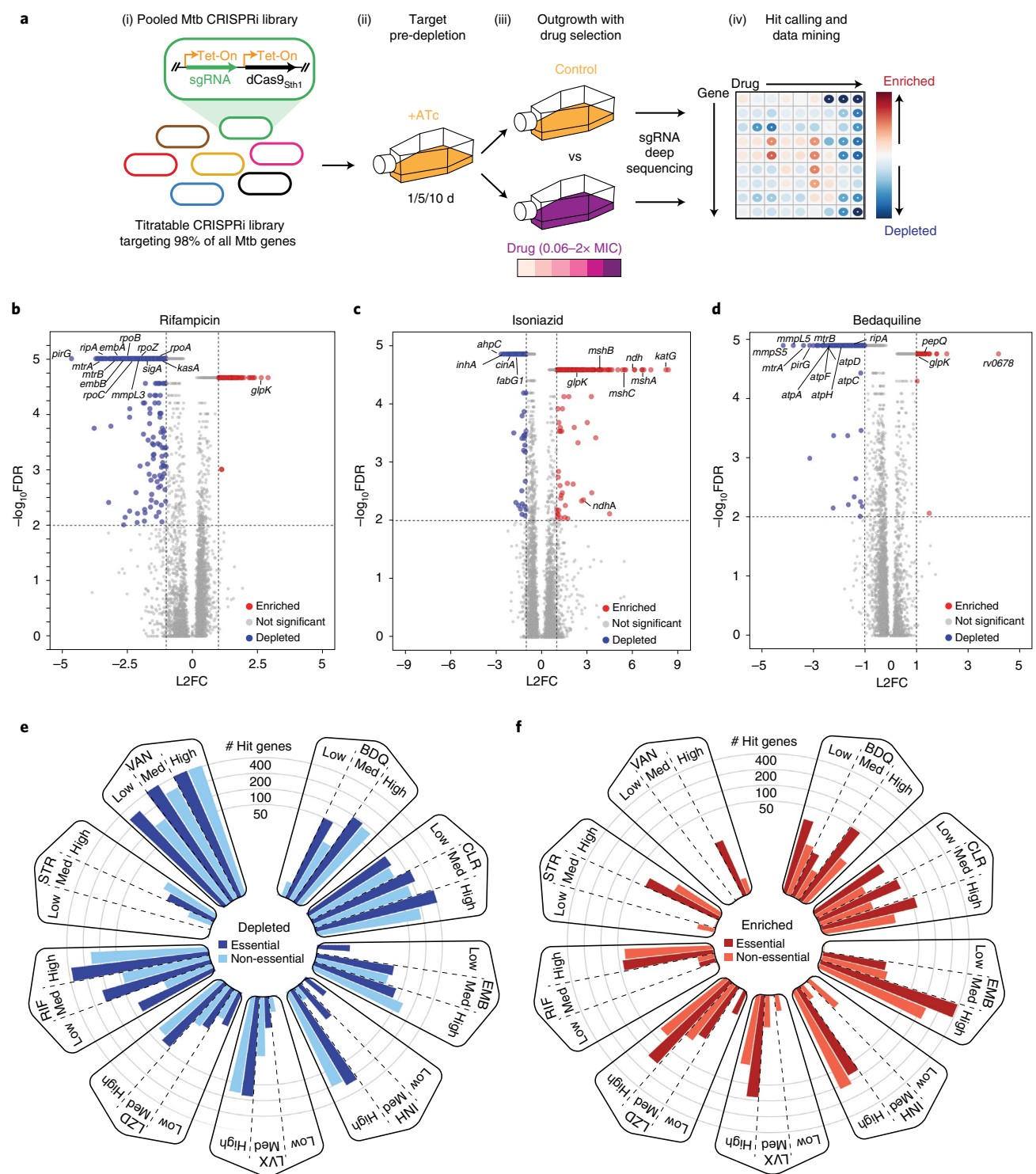

**Fig. 1 | Chemical-genetic profiling identifies hundreds of genes that influence drug efficacy in *M. tuberculosis*. a**, Quantifying chemical-genetic interactions in Mtb. (i) The pooled CRISPRi library contains 96,700 sgRNAs targeting 4,052/4,125 Mtb genes. In vitro essential genes[15] were targeted for titratable knockdown by varying the targeted PAM and sgRNA targeting sequence length; non-essential genes were targeted only with the strong sgRNAs. (ii) The CRISPRi inducer ATc was added for 1, 5 or 10 d before drug exposure to pre-deplete target gene products. (iii) Triplicate cultures were outgrown +ATc in DMSO or drug at concentrations spanning the predicted MIC. (iv) Following outgrowth, genomic DNA was collected from cultures treated with three descending doses of partially inhibitory drug concentrations ('High', 'Med' and 'Low'; Extended Data Fig. 1), sgRNAs amplified for deep sequencing, and hit genes called with MAGeCK. Growth phenotypes were highly correlated among triplicate screens (average Pearson correlation between replicate screens: *r* > 0.99). **b–d**, Volcano plots showing log₂ fold change (L2FC) values and false discovery rates (FDR) for each gene for the indicated drugs ('High' concentration, 1 d CRISPRi library pre-depletion for RIF and 5 d for INH and BDQ). **e,f**, The number of significantly depleted and enriched hit genes (FDR < 0.01, |L2FC| > 1) are shown for the indicated drugs. Hit genes were defined as the union of 1 and 5 d target pre-depletion screens because these datasets recovered the majority (>95%) of unique hits (Extended Data Fig. 2). Gene essentiality was defined by CRISPRi[15].

(single guide RNA) abundance by deep sequencing. Hit genes were identified by MAGeCK[18]. In total, we identified 1,373 genes whose knockdown led to sensitization and 775 genes whose knockdown led to resistance to at least one drug (Source Data Fig. 1). Most hit genes had a single chemical-genetic interaction, but some had as many as seven (Extended Data Fig. 2a).

The chemical-genetic screens recovered expected hit genes including direct drug targets, genes encoding the targets of known synergistic drug combinations and genes whose inactivation is known to confer acquired drug resistance (Fig. 1b–d)[3,8–10,19]. Benchmarking our CRISPRi approach against published transposon sequencing (TnSeq) chemical-genetic results revealed strong overlap (63.3–87.7% TnSeq hit recovery; Source Data Extended Data Fig. 3)[3], although TnSeq as currently implemented in Mtb is restricted to interrogation of non-essential genes. The number of hit genes varied widely across drugs (Fig. 1e,f and Extended Data Fig. 2b–j). Interestingly, essential genes were enriched relative to non-essential genes for chemical-genetic interactions (Fig. 1e,f and Extended Data Fig. 3a), even when taking into account the bias towards sgRNAs targeting essential genes in the CRISPRi library. This enrichment demonstrates the increased information content available when assaying essential genes by chemical genetics. Hierarchical clustering revealed unique chemical-genetic signatures for each drug (Extended Data Fig. 3b), which were then mined for biological insight.

Although they target distinct cellular processes, clustering analysis revealed correlated chemical-genetic signatures for rifampicin (RIF), vancomycin (VAN) and bedaquiline (BDQ), suggesting shared mechanisms of intrinsic resistance or sensitivity. Enrichment analysis identified the essential mycolic acid-arabinogalactan-peptidoglycan (mAGP) complex to be a common sensitizing hit for these drugs, but not the ribosome-targeting drugs clarithromycin (CLR), linezolid (LZD) or streptomycin (STR) (Extended Data Fig. 3c). The mAGP is the primary constituent of the Mtb cell envelope and has long been known to serve as a permeability barrier that mediates intrinsic drug resistance[6,7,20,21]. Interestingly, it was not obvious which drug physiochemical properties were driving selective mAGP sensitization (Extended Data Fig. 3d)[22]. For example, despite similar molecular weights, bedaquiline displayed a strong mAGP signature, whereas streptomycin did not; despite similar polar surface areas, rifampicin displayed a strong mAGP signature, but clarithromycin did not.

To ensure the validity of the screen results, we quantified drug susceptibility with individual hypomorphic CRISPRi strains targeting mAGP-biosynthetic genes, demonstrating 2- to 43-fold reductions in $IC_{50}$ (concentration required for 50% growth inhibition) for rifampicin, vancomycin and bedaquiline, but little to no change for linezolid (Extended Data Fig. 4a–d). To validate these results chemically, we focused on the β-ketoacyl-ACP synthase KasA, an enzyme essential for mycolic acid biosynthesis and an active drug target[23,24]. We confirmed that a small-molecule KasA inhibitor GSK3011724A (GSK'724A) synergizes with rifampicin, vancomycin and bedaquiline, but not linezolid, in laboratory culture and also confirmed

GSK'724A-rifampicin synergy ex vivo in macrophages (Extended Data Fig. 4e–g). We found that Mtb pre-treated with GSK'724A showed increased uptake of ethidium bromide and a fluorescent vancomycin conjugate, suggesting that the drug synergies observed with GSK'724A may be explained, at least in part, by the ability of GSK'724A to disrupt mAGP integrity and promote drug uptake (Extended Data Fig. 4h,i). These results validate the screen and confirm the role of the mAGP complex as a selective mechanism of intrinsic resistance relevant for some antitubercular agents but not others[25].

**mtrA promotes envelope integrity and intrinsic resistance.** Two of the most sensitizing hit genes across multiple drugs were *mtrA* and *mtrB* (Fig. 2a), which encode the response regulator MtrA and its cognate histidine kinase MtrB[26,27]. The *mtrAB* operon also encodes a putative lipoprotein *lpqB* (Fig. 2b). LpqB is proposed to interact with MtrB to promote MtrA phosphorylation and activation[28]. The similarities between the chemical-genetic signatures of *mtrAB-lpqB* and mAGP-biosynthetic genes (Fig. 2a and Extended Data Fig. 4b) suggest a role for this two-component system in regulating mAGP integrity. Given the predicted essentiality of *mtrAB-lpqB*[26,29] and the magnitude by which their inhibition sensitized Mtb to antibiotics, we next sought to better define the mechanism by which *mtrAB-lpqB* promote intrinsic drug resistance.

Strong CRISPRi silencing of *mtrA*, *mtrB* and *lpqB* prevented Mtb growth (Fig. 2b). Inhibition of *mtrA* was bacteriostatic in laboratory culture but bactericidal in macrophages (Fig. 2c and Extended Data Fig. 5a,b). *mtrA*, *mtrB* and *lpqB* knockdown strongly sensitized Mtb to rifampicin, vancomycin and bedaquiline (Fig. 2d and Extended Data Fig. 5c). As with KasA inhibition (Extended Data Fig. 4h,i), silencing *mtrA*, *mtrB*, and to a lesser extent *lpqB*, led to increased permeability to ethidium bromide and a fluorescent vancomycin conjugate (Fig. 2e), suggesting that the increase in drug susceptibility with *mtrAB-lpqB* knockdown is at least in part mediated by increased envelope permeability.

To better understand the mechanism(s) by which *mtrA* promotes intrinsic resistance, we next defined its regulon. RNA-seq following *mtrA* silencing identified 41 downregulated and 11 upregulated genes (Fig. 2f and Source Data Fig. 2), many of which were previously found to bind MtrA by ChIP-seq (Extended Data Fig. 5d,e)[27]. We confirmed by quantitative reverse transcription PCR (RT-qPCR) that putative regulon genes activated by MtrA were downregulated following *mtrAB* knockdown (Extended Data Fig. 5f). Surprisingly, silencing *lpqB* produced the opposite result—MtrA regulon genes were upregulated (Extended Data Fig. 5g). In contrast to the proposed role of LpqB as a positive regulator of this pathway[28], these data instead suggest that LpqB may negatively regulate MtrA signalling. To distinguish whether MtrA activation in the absence of LpqB is MtrB-dependent, we tested MtrA regulon expression upon simultaneous silencing of *mtrB* and *lpqB*. MtrA activation in the absence of LpqB required MtrB (Fig. 2g). Taken together, these results support a model whereby the extracytoplasmic lipoprotein LpqB functions as a negative regulator of MtrB to restrain MtrA activation.

**Fig. 2 | The response regulator MtrA promotes envelope integrity and intrinsic drug resistance. a**, Heatmap depicting chemical-genetic interactions from the 5 d CRISPRi library pre-depletion screen. The colour of each circle represents the gene-level L2FC. The white dot represents FDR < 0.01 and |L2FC| > 1. **b**, Growth was monitored by spotting serial dilutions of each strain on the indicated media. NT, non-targeting sgRNA; KD, knockdown; CR, CRISPRi-resistant. Transcriptional start sites[81] are indicated with black arrows. **c**, Growth (mean ± s.e.m., 3 biological replicates) of CRISPRi strains in IFN-γ-activated murine bone marrow-derived macrophages. Significance was determined by two-way analysis of variance (ANOVA) and adjusted for multiple comparisons. ****$P < 0.0001$. **d**, Dose-response curves (mean ± s.e.m., $n = 3$ biological replicates) for the indicated strains. **e**, Ethidium bromide and Vancomycin-BODIPY uptake (mean ± s.e.m., $n = 4$ biological replicates) of the indicated strains. Results from an unpaired *t*-test are shown; ****$P < 0.0001$. **f**, *mtrA* and NT CRISPRi strains were grown for 2 d with ATc, after which RNA was collected and sequenced. Dashed lines mark significant hit genes ($-\log_{10}(P_{adj}) < 0.05$ and |L2FC| > 1). **g**, Quantification (mean ± s.e.m., $n = 3$ biological replicates) of gene mRNA levels by RT-qPCR. Strains were grown ±ATc for ~3 generations before collecting RNA. Results from an unpaired *t*-test are shown; **$P < 0.01$, ***$P < 0.001$, ****$P < 0.0001$. **h**, Schematic of the proposed MtrAB-LpqB signalling system. Created with BioRender.com.

The MtrA regulon encodes numerous peptidoglycan remodelling enzymes important for cell growth and division (Source Data Fig. 2)[27,30,31]. Intriguingly, several regulon genes were also identified as sensitizing hits in our screen, phenocopying the effects of *mtrA* silencing (Fig. 2a). *mtrAB* and regulon expression were not altered in response to drug treatment (Extended Data Fig. 5h).

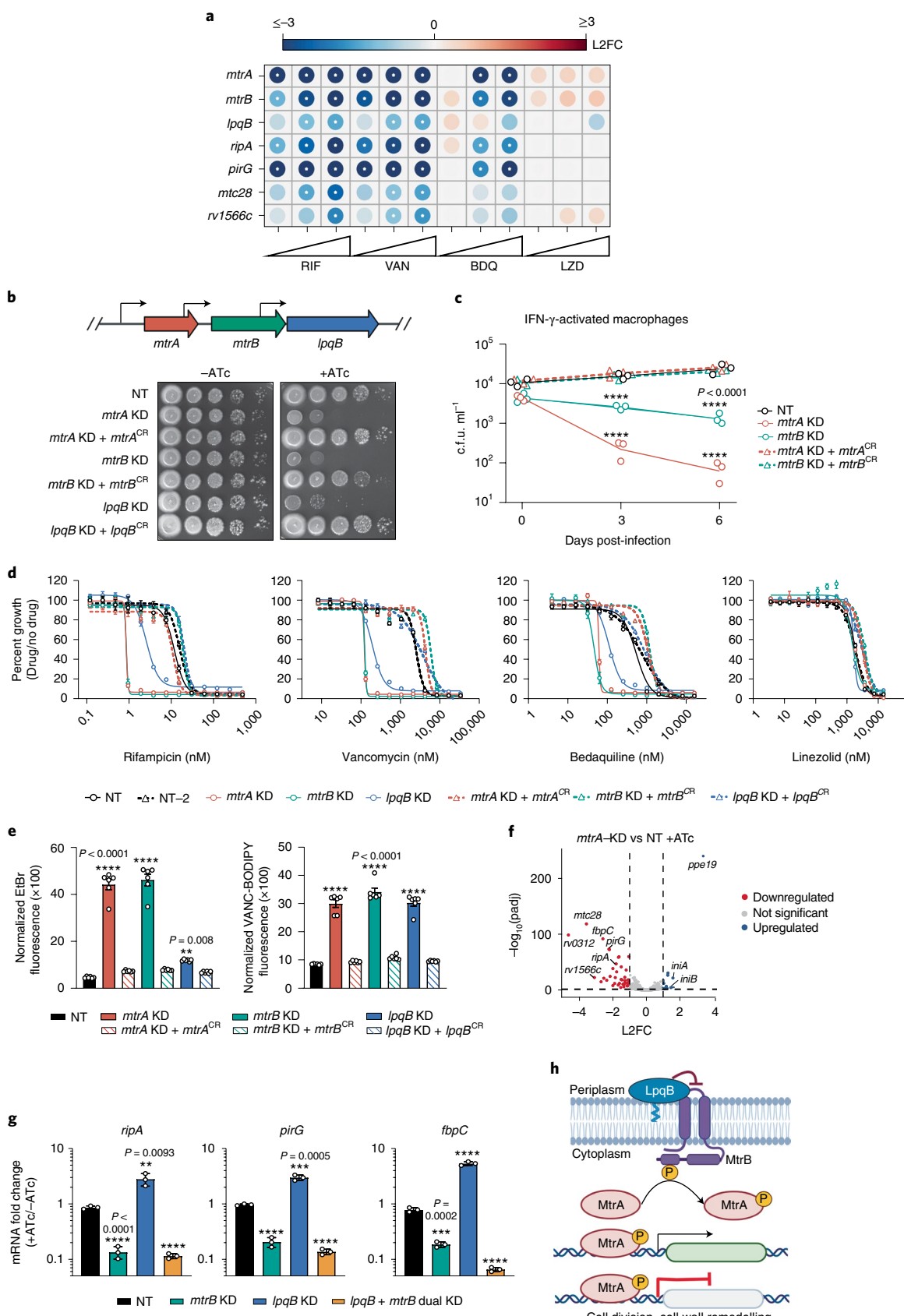

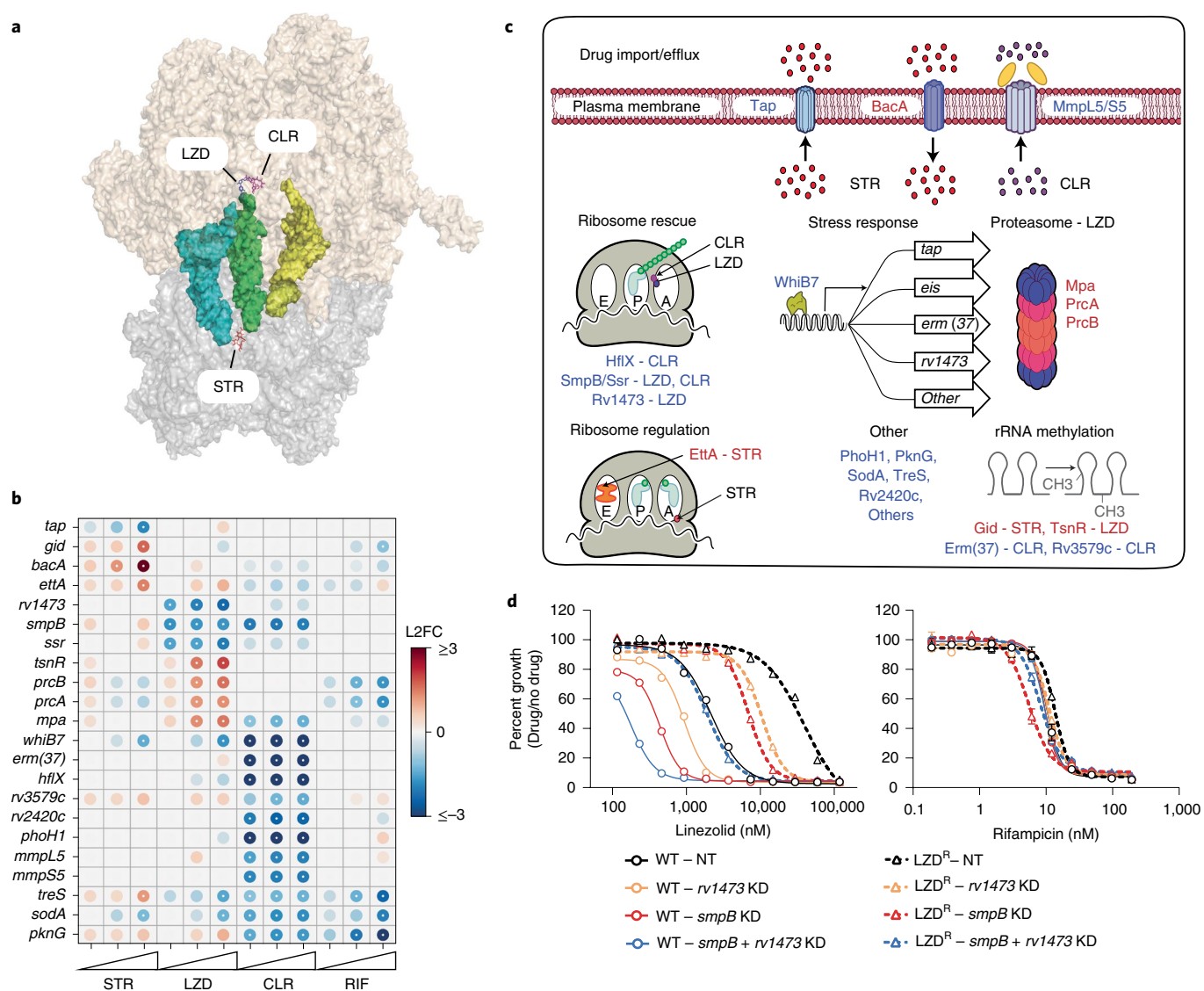

**Fig. 3 | Diverse pathways contribute to intrinsic resistance and susceptibility to ribosome-targeting antibiotics. a**, Structure of LZD, CLR and STR bound to the *Thermus thermophilus* ribosome. PDB codes: 3DLL, 1J5A, 1FJG, 4V5C. **b**, Heatmap depicting chemical-genetic interactions as in Fig. 2a. **c**, Chemical-genetic hit genes from Fig. 3b are involved in diverse cellular pathways. Genes whose inhibition decreased or increased fitness in the indicated drug are listed in blue or red, respectively. **d**, Dose-response curves (mean ± s.e.m., *n* = 3 biological replicates) for the indicated CRISPRi strains in H37Rv or *rplC*-Cys154Arg linezolid-resistant H37Rv (LZD^R).

These results demonstrate the central role of the MtrAB signalling pathway in coordinating proper peptidoglycan remodelling during bacterial growth and division (Fig. 2h), and highlight the potential utility of small-molecule modulators of this pathway.

**Diverse pathways influence potency of translation inhibitors.** Inhibition of mAGP biosynthesis did not sensitize Mtb to the three ribosome-targeting drugs streptomycin, clarithromycin and linezolid (Extended Data Figs. 3c,d and 6a). Thus, mAGP inhibition is unlikely to be a relevant mechanism to potentiate activity of these drugs.

Streptomycin, clarithromycin and linezolid had correlated chemical-genetic signatures that appeared to be driven by the lack of a mAGP signature rather than any unique ribosome target signature (Extended Data Fig. 3b,c), which may reflect the different mechanisms of action of these drugs. Streptomycin interacts with the 30S ribosomal subunit and induces mis-translation[32], whereas

linezolid and clarithromycin both act on the 50S subunit to inhibit translation elongation[33,34] (Fig. 3a). The sole sensitizing hit observed uniquely among the three ribosome-targeting drugs was *whiB7*, a transcription factor that induces a stress response promoting intrinsic resistance to numerous ribosome-targeting antibiotics[35]. The specific hit genes varied between these drugs but could be broadly classified into the following categories: ribosomal RNA (rRNA) methylation, drug efflux/import, ribosome rescue, ribosome regulation, proteasome activity and numerous poorly characterized genes (Fig. 3b,c).

Consistent with previous publications, we found that rRNA methyltransferases can confer either intrinsic sensitivity or intrinsic resistance to ribosome-targeting drugs[36]. For example, silencing *erm(37)* resulted in clarithromycin sensitivity (Fig. 3b)[37], whereas silencing the 16S rRNA methyltransferase *gid* conferred streptomycin resistance[38]. Interestingly, knockdown of the predicted 23S rRNA methyltransferase *tsnR* conferred resistance to linezolid (Fig. 3b).

 

This is analogous to work in *Staphylococcus aureus*, in which loss of the evolutionarily distinct 23S methyltransferase *rlmN* confers linezolid resistance both in vitro and in the clinic[39]. To determine whether loss-of-function *tsnR* mutations could play a clinically relevant role in acquired linezolid resistance, we assembled a database of >45,000 whole genome sequences from Mtb clinical isolates (Source Data Extended Data Fig. 8). Given the recent introduction of linezolid to TB treatment, we expected linezolid-resistant Mtb to be rare. Consistent with this, we identified the most common linezolid-resistance mutation *rplC*-Cys154Arg[40] only 122 times in our genome database. While putative loss-of-function mutations in *tsnR* were even more rare (Source Data Fig. 3), two multidrug-resistant (MDR) Mtb strains harboured both a *tsnR* frameshift allele and an *rplC*-Cys154Arg allele, which are highly unlikely to have co-occurred by chance ($\chi^2$ test with Yates' correction: $P < 0.0001$). These data highlight that loss-of-function *tsnR* mutations may serve as stepping stones to high-level resistance as linezolid is used more widely in the clinic.

We next sought to determine whether our findings could be exploited to identify synergistic drug–target combinations to overcome resistance and increase the therapeutic index for linezolid, analogous to preclinical efforts to boost ethionamide potency and tolerability[41]. The essential trans-translation genes, *smpB* and *ssr*[29], were identified as linezolid-sensitizing hits (Fig. 3b,c). Trans-translation is a ribosome rescue pathway thought to primarily rescue ribosomes stalled on non-stop messenger RNAs[42]. Additionally, we identified the poorly characterized gene *rv1473* as a linezolid-specific sensitizing hit (Fig. 3b,c). *rv1473* was previously reported to be a macrolide efflux pump[43]. However, the lack of predicted transmembrane helices suggest that *rv1473* is unlikely to be a membrane-embedded ABC transporter. Rather, homology suggests that *rv1473* belongs to the family of antibiotic resistance ABC-F proteins, which have been shown to rescue drug-bound ribosomes by facilitating drug dissociation (Extended Data Fig. 6b)[44,45].

Our data suggest that inhibition of *rv1473* and trans-translation could make linezolid more potent. Individual CRISPRi knockdown of *rv1473* and *smpB* lowered the IC$_{50}$ for linezolid by 2.3- and 5-fold, respectively (Fig. 3d and Extended Data Fig. 6c). Inhibition of the Clp protease did not sensitize Mtb to linezolid (Extended Data Fig. 6d,e), consistent with the critical role of trans-translation in rescuing linezolid-stalled ribosomes, but not in Clp protease-mediated turnover of *ssrA*-tagged stalled translation products[46]. Dual CRISPRi knockdown of both *rv1473* and *smpB* lowered the linezolid IC$_{50}$ by 12.2-fold (Fig. 3d and Extended Data Fig. 6c), consistent with *rv1473* and trans-translation functioning in separate intrinsic resistance pathways. Given the magnitude of linezolid sensitivity of the dual knockdown strain, we hypothesized that dual inhibition of *rv1473* and *smpB* could functionally reverse linezolid resistance. Dual knockdown of *rv1473* and *smpB* in a linezolid-resistant strain restored linezolid sensitivity back to wild-type (WT) levels (Fig. 3d and Extended Data Fig. 6f,g), demonstrating that inhibition of intrinsic resistance factors can potentiate linezolid and functionally reverse acquired drug resistance.

**bacA mutations are a source of aminoglycoside resistance.**
Acquired drug resistance is a major barrier to successful TB treatment. However, our knowledge of the genetic basis of acquired drug resistance remains incomplete, particularly for mutations outside of the drug target or activator, and which typically confer low-to-intermediate, but clinically relevant, levels of drug resistance[4,8,11,12]. Given the ability of our chemical-genetic approach to identify hit genes associated with clinically relevant acquired drug resistance (Figs. 1b–d and 3b), we hypothesized that mining our chemical-genetic data may identify previously unrecognized mechanisms of acquired drug resistance in Mtb.

We chose to focus our search for sources of acquired drug resistance to streptomycin. Streptomycin has been used to treat TB since the late 1940s and thus Mtb has had decades of selective pressure to potentially give rise to a diverse set of resistance mutations. Aminoglycosides such as streptomycin must traverse the Mtb envelope to access their ribosomal target. The mechanism(s) of aminoglycoside uptake by Mtb are not well understood. Interestingly, and consistent with previous work[47], the strongest streptomycin-resistance hit gene in our screen was *bacA* (*rv1819c*; Fig. 3b,c). Recently, structural and biochemical work demonstrated that *bacA* is an ABC importer of diverse hydrophilic solutes[48]. Thus, we hypothesized that *bacA* may serve as a streptomycin importer and that *bacA* loss-of-function mutants may be an unrecognized source of streptomycin resistance in clinical Mtb strains.

Searching our clinical strain genome database, we observed numerous *bacA* non-synonymous single nucleotide polymorphisms (SNPs) and small insertion-deletions (indels), and selected six for experimental validation (Fig. 4a and Source Data Fig. 4). Consistent with our hypothesis, four of the six *bacA* mutant strains displayed an elevated streptomycin MIC (Fig. 4b). Interestingly, these strains also showed elevated MICs to other aminoglycosides (amikacin and kanamycin) as well as the tuberactinomycin capreomycin (Extended Data Fig. 7a). Moreover, overexpression of Mtb *bacA* sensitized *M. smegmatis* to streptomycin (Extended Data Fig. 7b). While further studies are necessary to definitively show that *bacA* is an importer of aminoglycosides and tuberactinomycins, our data, in combination with previous studies[47,48], strongly suggest that BacA imports these hydrophilic drugs (Extended Data Fig. 7c). Since a Δ*bacA* strain is not entirely resistant to aminoglycosides and tuberactinomycins, other import mechanisms must exist.

**ettA mutations confer low-level resistance to diverse drugs.**
Another strong streptomycin-resistance hit in our screen was *rv2477c*, which also showed low-level resistance to other drugs (Fig. 3b,c). *rv2477c* is an orthologue of the *Escherichia coli* gene *ettA* (~58% amino acid identity), a ribosome-associated ABC-F protein that regulates the translation elongation cycle[49–51]. Due to its sequence similarity, we will refer to *rv2477c* as *ettA*. Biochemical studies demonstrated that ATP-bound EttA from *E. coli* stimulates formation of the first peptide bond of the initiating ribosome and then, concomitant with ATP hydrolysis, dissociates from the ribosome to allow translation elongation[49,50]. Unlike *ettA* in *E. coli*, *ettA* is essential in Mtb[15].

Using our clinical strain genome database, we selected four non-synonymous *ettA* SNPs for experimental validation (Fig. 5a and Source Data Fig. 5). We also included an *ettA*-Trp135Gly mutation that was identified in serial Mtb isolates from a patient in Thailand directly preceding the transition from MDR-TB to extensively drug-resistant (XDR) TB[52]. The *ettA*-Trp135Gly isolate became phenotypically resistant to kanamycin and amikacin, but harboured no known resistance mutations to either aminoglycoside. All candidate SNPs were capable of complementing knockdown of the endogenous *ettA* allele (Fig. 5b). Both the Gly41Glu and Trp135Gly variants displayed a modest growth defect, suggesting that these two SNPs are partial loss-of-function mutations (Extended Data Fig. 7d,e). Consistent with a role for these *ettA* SNPs in conferring acquired drug resistance, four out of the five variants showed an increased MIC for streptomycin (Fig. 5c). The Gly41Glu and Trp135Gly strains showed >5-fold shifts in streptomycin IC$_{50}$, similar in magnitude to clinically relevant *gid* mutants[38], as well as low-level resistance to a mechanistically diverse panel of antibiotics including amikacin, ethambutol (EMB), rifampicin and levofloxacin (LVX) (Fig. 5c and Extended Data Fig. 7f,g).

To determine the mechanism by which *ettA* SNPs may confer low-level acquired multidrug resistance, we analysed the *M. smegmatis* proteome after silencing the *ettA* homologue, *ms4700*[15].

 

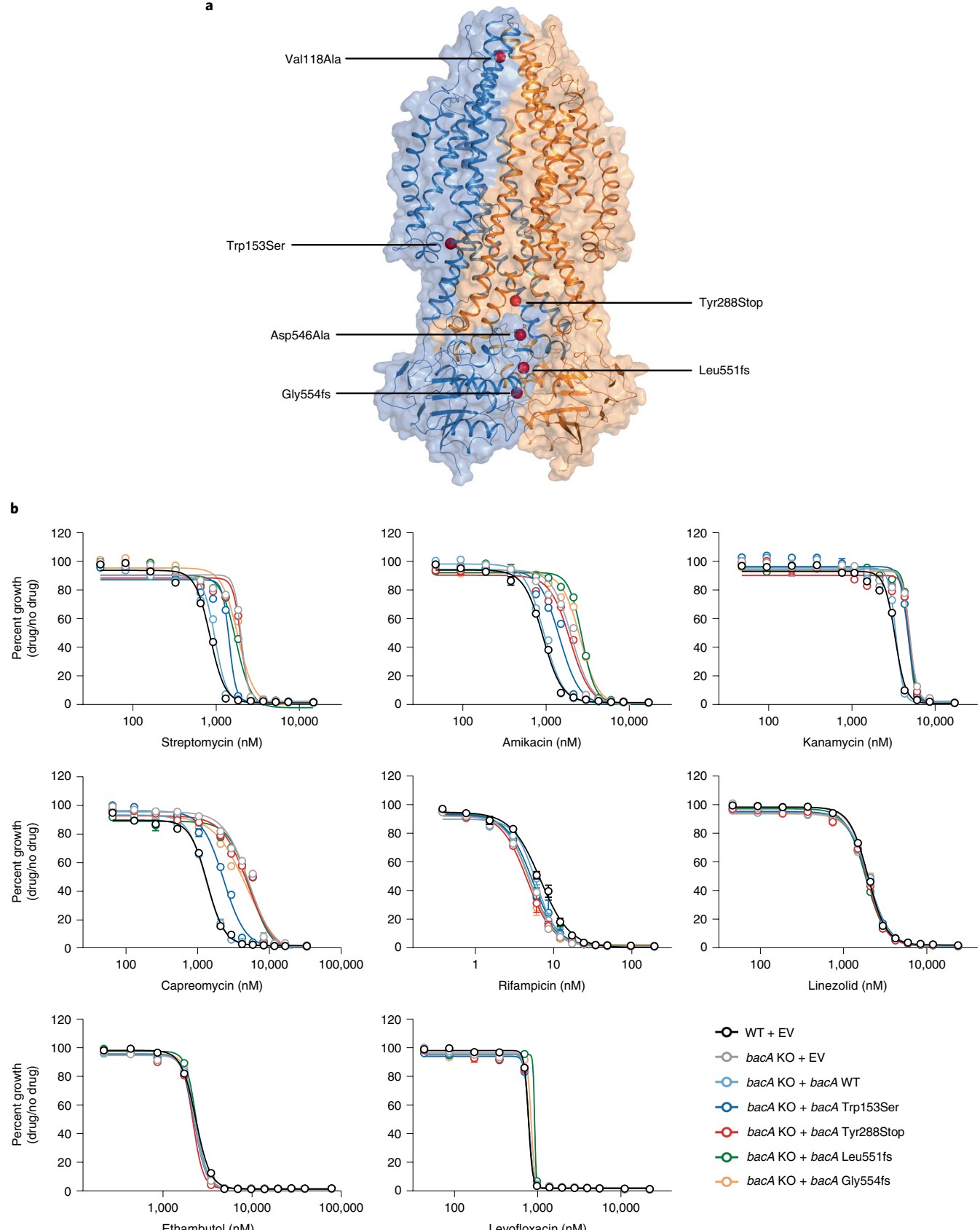

**Fig. 4 | Loss-of-function mutations in *bacA* confer acquired drug resistance to aminoglycosides and capreomycin. a**, BacA structure (PDB: 6TQF). Red spheres mark sites of experimentally tested *bacA* SNPs and frameshift (fs) causing indels from clinical Mtb strains. **b**, Dose-response curves (mean ± s.e.m., $n = 3$ biological replicates) of Mtb strains harbouring *bacA* mutations. KO, knockout; EV, empty complementation vector.

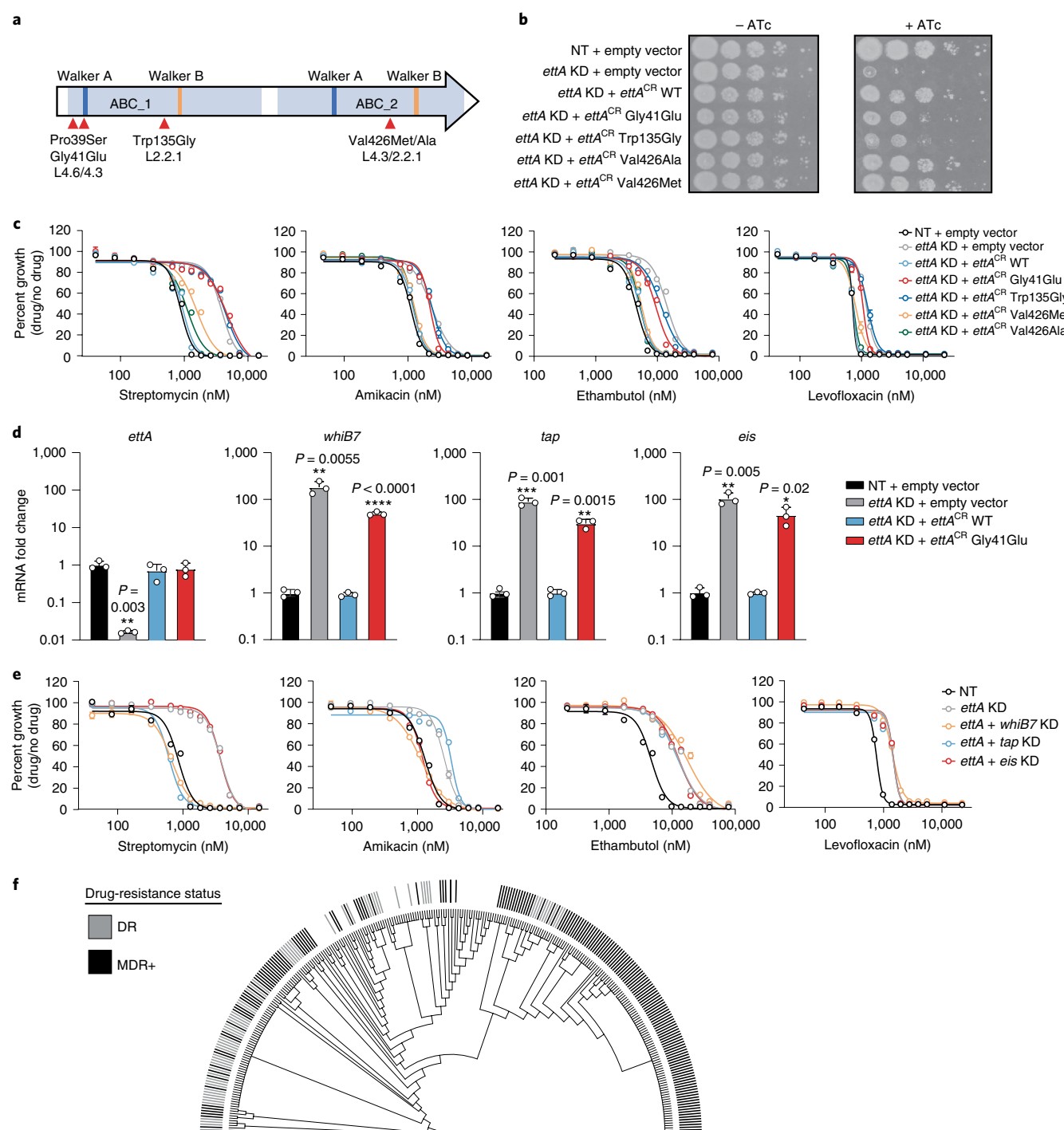

**Fig. 5 | Mutations in the translation factor EttA constitutively upregulate the *whiB7* stress response and confer low-level, acquired multidrug resistance.**
**a**, EttA domain organization. ABC domains, Walker A and Walker B motifs are highlighted in light blue, dark blue and orange, respectively. SNPs tested are highlighted with red arrows. **b**, Growth was monitored by spotting serial dilutions of each strain on the indicated media. The *ettA* CRISPRi strain was complemented with an empty vector or CRISPRi-resistant alleles harbouring the indicated *ettA* SNPs. **c**, Dose-response curves (mean ± s.e.m., *n* = 3 biological replicates) for strains harbouring *ettA* SNPs. **d**, Quantification (mean ± s.e.m., *n* = 3 biological replicates) of gene mRNA levels by RT-qPCR. Strains were grown +ATc for ~5 generations before collecting RNA. Statistical significance was calculated with Student's *t*-test; *$P < 0.05$, **$P < 0.01$, ***$P < 0.001$, ****$P < 0.0001$. **e**, Dose-response curves (mean ± s.e.m., *n* = 3 biological replicates) for *ettA* single and dual knockdown strains. **f**, Phylogenetic tree of 291 Mtb clinical strains harbouring the *ettA*-Gly41Glu variant (Source Data Fig. 5). Genotypically predicted drug-resistance status is shown. DR, resistance-conferring mutations to rifampicin, isoniazid, pyrazinamide or ethambutol present; MDR+, resistance-conferring mutations to a minimum of rifampicin and isoniazid.

Two of the most upregulated proteins upon *ms4700* knockdown were HflX and Eis (Extended Data Fig. 7h), known members of the *whiB7* regulon in Mtb[35]. Thus, we hypothesized that partial

loss-of-function *ettA* alleles may promote drug resistance by stalling translation and constitutively upregulating the *whiB7* stress response, in essence mimicking the effects of translation stress caused by

ribosome inhibitors to activate *whiB7*[53]. Consistent with this hypothesis, *whiB7* and regulon genes[35] were constitutively upregulated in the *ettA*-Gly41Glu Mtb mutant (Fig. 5d). Furthermore, simultaneous knockdown of *ettA* and *whiB7* was able to specifically reverse aminoglycoside resistance (Fig. 5e and Extended Data Fig. 7i). *whiB7* knockdown did not reverse ethambutol or levofloxacin resistance, suggesting that the mechanism by which *ettA* mutations confer resistance to these drugs is *whiB7*-independent and may instead be due to changes in translation and/or growth rates[54,55].

Further epidemiological analysis focused on *ettA*-Gly41Glu, the most common *ettA* SNP in our database (*n* = 291, ~0.7% of all Mtb strains). Phylogenetic analysis shows that this cluster of related strains is heavily enriched for additional acquired drug resistance mutations (Fig. 5f and Source Data Fig. 5). *ettA*-Gly41Glu strains are found in multiple countries but are concentrated in Peru[56] and indigenous communities of Colombia[57], where they are associated with an MDR-TB outbreak (Fig. 5f).

**A *whiB7* mutation renders an Mtb sublineage sensitive to CLR.** Since *ettA* mutations appear to confer resistance by constitutive *whiB7* activation, we next mined our clinical strain genome database to identify putative gain-of-function *whiB7* mutations that may be associated with acquired drug resistance[58]. We identified numerous putative gain-of-function mutations in the *whiB7* promoter, 5'UTR, and upstream open reading frame (uORF), most of which have not been previously recognized as potential acquired drug resistance determinants (Fig. 6a and Source Data Fig. 6)[58–60]. Unexpectedly, the most common *whiB7* variant in our database was a putative loss-of-function allele. This allele, whiB7 Gly64delG, represents nearly one-third (*n* = 851/3,186) of all *whiB7* variants in our database (Fig. 6a)[61–63]. Gly64delG results in a premature stop codon and truncation of the critical DNA binding AT-hook element (Extended Data Fig. 8a)[64], thus presumably inactivating WhiB7. *whiB7*-mediated intrinsic drug resistance typically renders macrolides ineffective in treating TB. We next sought to test whether this common Gly64delG mutation could render this subset of Mtb strains hypersusceptible to and treatable with macrolides.

Lineage calling identified the Gly64delG indel as uniquely present in all lineage 1.2.1 (L1.2.1) Mtb isolates, a major sublineage of the L1 Indo-Oceanic clade (Fig. 6a, Extended Data Fig. 8b and Source Data Fig. 6)[65]. Using a reference set of Mtb clinical strains[66], we first confirmed the presence of the *whiB7* Gly64delG indel in L1.2.1 (Fig. 6b). Consistent with loss of *whiB7* function, the L1.2.1 isolate was hypersusceptible to clarithromycin as well as other macrolides, ketolides and lincosamides, whereas all other clinical isolates were intrinsically resistant (Fig. 6c and Extended Data Fig. 8c–e). The *whiB7* Gly64delG allele failed to complement intrinsic clarithromycin resistance in an H37Rv Δ*whiB7* strain, confirming that Gly64delG is a loss-of-function allele (Extended Data Fig. 8f). We next tested the efficacy of clarithromycin against L1.2.1 in a low-dose aerosol mouse infection model, with drug dosing designed to mimic human pharmacokinetics[67] (Extended Data Fig. 9a–d). Consistent with the in vitro results, L1.2.1 but not control strains was susceptible to clarithromycin treatment in mice (Fig.6d,e and Extended Data Fig. 9e–k)[68,69].

To estimate the potential clinical impact of this finding, we next examined the geographic distribution of L1.2.1. This sublineage is found predominantly in Southeast Asia[65] and is highly prevalent in the Philippines, accounting for approximately 80% of all Mtb isolates in this country[70]. The Philippines has one of the highest TB incidence rates in the world, including a high burden of drug-resistant TB, and TB is a leading cause of death in this country[71]. L1.2.1 is estimated to cause ~600,000 cases of active TB per year globally[65], of which ~43,000 are estimated to be MDR-TB on the basis of the frequencies of drug resistance in our database. Thus, clarithromycin,

an effective, orally available, safe and generic antibiotic, could potentially be repurposed to treat this major Mtb sublineage.

## Discussion
A deeper understanding of the bacterial pathways that influence drug efficacy is needed to guide TB drug development and treatment. To address this challenge, we developed a CRISPRi platform to define the determinants that influence bacterial fitness in the presence of antibiotics, and then overlayed these chemical-genetic results with comparative genomics of Mtb clinical isolates. This work builds on previous chemical-genetic efforts in Mtb[3,72]. Caveats include that genetic and pharmacologic target inhibition are not the same[73], highlighting the importance of validating chemical-genetic interactions with small-molecule inhibitors. Moreover, all pooled screens may miss effects where the phenotype can be complemented in trans (for example cross-feeding)[74]. Despite these limitations, we illustrate the power of this dataset to derive clinically relevant biological insight. We uncover diverse mechanisms of intrinsic drug resistance that can be targeted to potentiate therapy, describe previously unknown mechanisms of acquired drug resistance, and make the unexpected discovery of an 'acquired drug sensitivity' that could enable the repurposing of clarithromycin to treat an Mtb sublineage.

We identified hundreds of genes that contribute to intrinsic drug resistance in Mtb that could serve as a foundation for the rational development of drug combinations to disarm intrinsic drug resistance. Our results confirm that one of the richest sources of potentially synergistic targets is the mycobacterial envelope[3,6,7]. Beyond the Mtb cell envelope, our results uncover both shared and unique intrinsic resistance and sensitivity mechanisms, as highlighted by profiling three ribosome-targeting antibiotics. Our results suggest that Rv1473 serves as an antibiotic resistance ABC-F protein, capable of displacing oxazolidinones and phenicols from the Mtb ribosome (Extended Data Fig. 6). Next-generation oxazolidinone analogues that are recalcitrant to such drug-displacing activity could be designed, analogous to third-generation tetracycline analogues which are resistant to the drug-displacing activity of the ABC-F protein TetM[75]. In light of our findings, we suggest designating *rv1473* as *ocrA* (oxazolidinone chloramphenicol resistance A). Finally, these results show that trans-translation may be a target for synergistic drug combinations[42], which could be important in increasing the potency and decreasing toxicity of oxazolidinones.

Discovery of previously unrecognized acquired drug resistance mechanisms will improve TB therapy by informing more effective molecular diagnostics and personalized treatment regimens. Our results illustrate the utility of combining chemical genetics with comparative genomics to power this discovery. First, we show that inhibition of *tsnR* increases Mtb fitness in the presence of linezolid, and thus mutations in this gene could be monitored as linezolid use is expanded in the clinic. Second, we find that loss-of-function mutations in the ABC importer *bacA* confer acquired drug resistance to four second-line TB drugs. Third, we show that partial loss-of-function mutations in *ettA* result in constitutive activation of the *whiB7* stress response and low-level acquired multidrug resistance. Low-level drug resistance has been associated with TB treatment failure[4], and could serve as a stepping stone to allow additional, high-level drug resistance mutations to evolve[76]. Our results provide a mechanistic explanation for previous findings[52], whereby serial Mtb isolates from a TB patient acquired a Trp135Gly mutation in *ettA* directly preceding the transition from MDR to XDR-TB. Of all Mtb strains in our genome database, 3.1% harbour a missense SNP in *ettA*, suggesting that *ettA*-mediated acquired drug resistance could be common. Capreomycin remains effective against *ettA* variants and should be considered for treatment. Lastly, we identified numerous additional genes as candidates for mechanisms of acquired drug resistance in Mtb (Source Data Extended Fig. 7), although further work is necessary to validate these predictions.

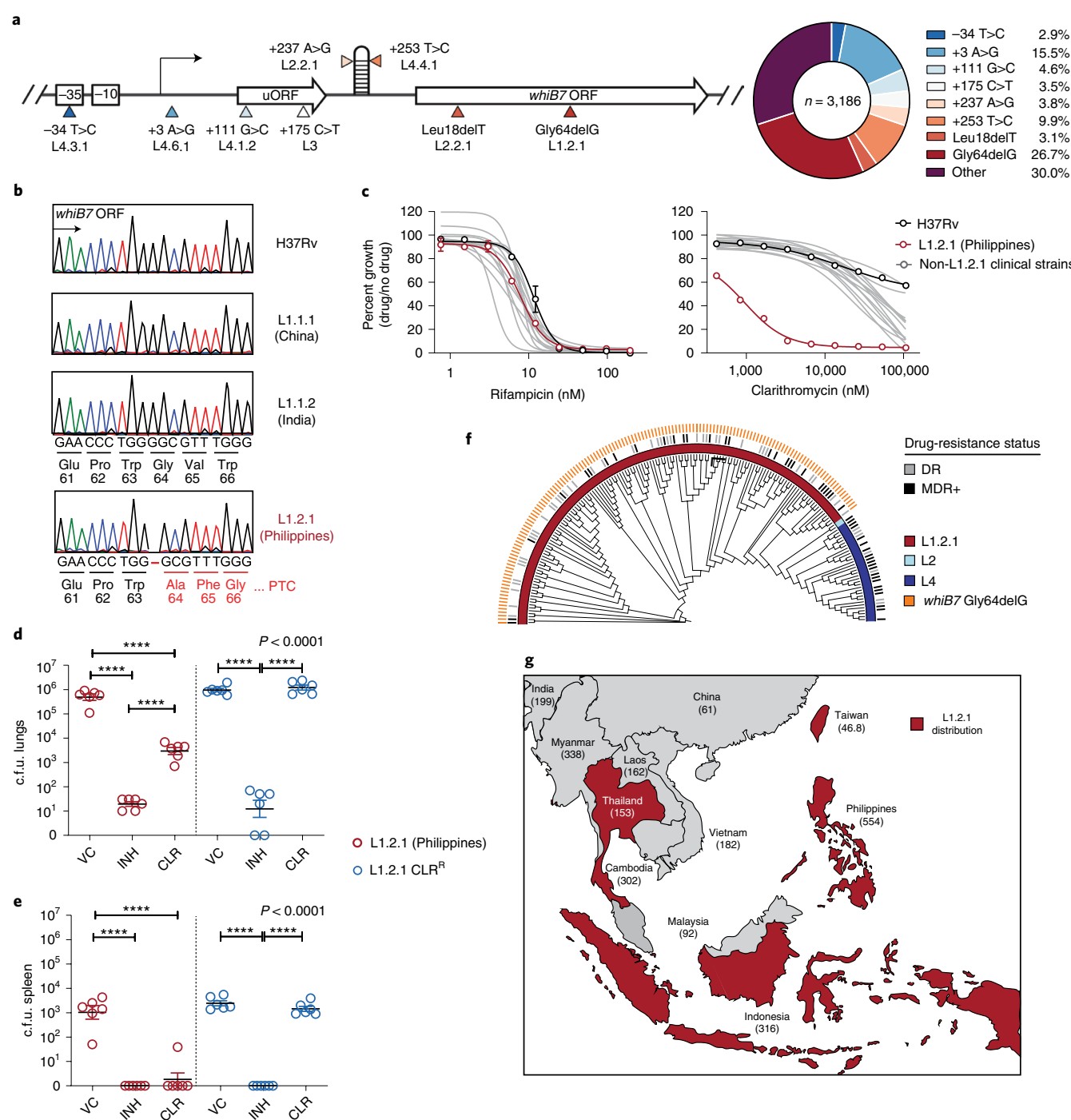

**Fig. 6 | A loss-of-function mutation in *whiB7* renders an endemic Indo-Oceanic Mtb lineage hypersusceptible to macrolides. a**, Diagram of Mtb *whiB7* with the eight most common *whiB7* variants observed in our clinical strain genome database. Pie chart depicts the observed frequencies of each variant. L, dominant lineage in which variant is observed. **b**, Sanger sequencing of *whiB7* from the indicated Mtb clinical strains and their country of origin. PTC, premature termination codon. The colour of each peak represents the base at the indicated position (black, G; green, A; red, T; blue, C). **c**, Dose-response curves (mean ± s.e.m., *n* = 3 biological replicates) were measured for a reference set of Mtb clinical and lab strains. **d,e**, Lung (**d**) and spleen (**e**) Mtb c.f.u. (mean ± s.e.m.) in BALB/c mice after 24 d of INH (25 mg kg⁻¹) or CLR (200 mg kg⁻¹) treatment. Statistical significance was assessed by one-way ANOVA followed by Tukey's post-hoc test. VC, vehicle control; CLRᴿ, clarithromycin-resistant (23S rRNA A2297G). Black line, median. *n* = 6 mice per group/condition. **f**, Phylogenetic tree of 178 Mtb clinical strains isolated during the 2012 nationwide drug resistance survey in the Philippines[70] (Source Data Fig. 6). The presence of the *whiB7* Gly64delG mutation and genotypically predicted drug-resistance status are shown as in Fig. 5f. **g**, Map showing L1.2.1 distribution in Southeast Asia and TB incidence rates of each country[71].

In the search for gain-of-function *whiB7* mutations that confer acquired drug resistance, we discovered common loss-of-function *whiB7* alleles, a phenomenon which we refer to as 'acquired drug sensitivity.' Beyond *whiB7*, we identified several additional candidate acquired drug sensitivity alleles (Supplementary Table 2) and validate loss-of-function *mmpL5* alleles in L1 and L7 isolates that

render them hypersusceptible to bedaquiline and the leprosy drug clofazimine (Extended Data Fig. 10)[11]. Phylogenetic dating suggests that the *whiB7* mutation arose approximately 900 years ago, well before the introduction of TB chemotherapy[77]. Since macrolides have not historically been used to treat TB, there has probably been little selective pressure against *whiB7* loss-of-function mutants. It remains unclear whether the Gly64delG mutation provides or provided a selective benefit to L1.2.1, enriched as a passenger mutation, or arose as a result of genetic drift. Regardless, we find that the entire L1.2.1 sublineage is a *whiB7* loss-of-function mutant which renders it susceptible to macrolides. These results pave the way for further preclinical efficacy studies to support that clarithromycin can be repurposed to treat this major Mtb sublineage.

## Methods

**Bacterial strains.** Mtb strains are derivatives of H37Rv unless otherwise noted. A reference set of Mtb clinical strains was obtained from the Belgian Coordinated Collections of Microorganisms (BCCM)[66]. *E. coli* strains are derivatives of DH5alpha (NEB).

**Mycobacterial cultures.** Mtb was grown at 37 °C in Difco Middlebrook 7H9 broth or on 7H10 agar supplemented with 0.2% glycerol (7H9) or 0.5% glycerol (7H10), 0.05% Tween-80, 1X oleic acid-albumin-dextrose-catalase (OADC) and the appropriate antibiotics (kanamycin 10–20 µg ml$^{-1}$ and/or zeocin 20 µg ml$^{-1}$). Anhydrotetracycline (ATc) was used at 100 ng ml$^{-1}$. Mtb cultures were grown standing in tissue culture flasks (unless otherwise indicated) with 5% $CO_2$.

**Generation of individual CRISPRi and CRISPRi-resistant complementation strains.** Individual CRISPRi plasmids were cloned as described in ref. [78] using Addgene plasmid 166886. Briefly, the CRISPRi plasmid backbone was digested with BsmBI-v2 (NEB R0739L) and gel purified. sgRNAs were designed to target the non-template strand of the target gene ORF. For each individual sgRNA, two complementary oligonucleotides with appropriate sticky end overhangs were annealed and ligated (T4 ligase NEB M0202M) into the BsmBI-digested plasmid backbone. Successful cloning was confirmed by Sanger sequencing.

Individual CRISPRi plasmids were then electroporated into Mtb. Electrocompetent cells were obtained as described in ref. [79]. Briefly, an Mtb culture was expanded to an $OD_{600}$ = 0.8–1.0 and pelleted (4,000 × g for 10 min). The cell pellet was washed three times in sterile 10% glycerol. The washed bacilli were then resuspended in 10% glycerol in a final volume of 5% of the original culture volume. For each transformation, 100 ng plasmid DNA and 100 µl electrocompetent mycobacteria were mixed and transferred to a 2 mm electroporation cuvette (Bio-Rad 1652082). Where necessary, 100 ng plasmid plRL19 (Addgene plasmid 163634) was also added. Electroporation was performed using the Gene Pulser X cell electroporation system (Bio-Rad 1652660) set at 2,500 V, 700 Ω and 25 µF. Bacteria were recovered in 7H9 for 24 h. After the recovery incubation, cells were plated on 7H10 agar supplemented with the appropriate antibiotic to select for transformants.

To complement CRISPRi-mediated gene knockdown, synonymous mutations were introduced into the complementing allele at both the protospacer adjacent motif (PAM) and the seed sequence (the 8–10 PAM-proximal bases at the 3' end of the sgRNA targeting sequence) to prevent sgRNA targeting. Silent mutations were introduced into Gibson assembly oligos to generate 'CRISPRi-resistant' alleles. Complementation alleles were expressed from the endogenous or *hsp60* promoters in a Tweety or Giles integrating plasmid backbone, as indicated in each figure legend and/or the plasmid descriptions. These alleles were then transformed into the corresponding CRISPRi knockdown strain, with the plRL40 Giles Int expressing plasmid where necessary. The full list of sgRNA targeting sequences and complementation plasmids can be found in Supplementary Table 2.

**Pooled CRISPRi chemical-genetic screening.** Chemical-genetic screens were initiated by thawing 5 × 1 ml (1 $OD_{600}$ unit per ml) aliquots of the Mtb CRISPRi library[15] (RLC12; Addgene 163954) and inoculating each aliquot into 19 ml 7H9 supplemented with kanamycin (10 µg ml$^{-1}$) in a vented tissue culture flask (T-75; Falcon 353136). The starting $OD_{600}$ of each culture was approximately 0.05. Cultures were expanded to $OD_{600}$ = 1.5, pooled and passed through a 10 µm cell strainer (pluriSelect 43-50010-03) to obtain a single-cell suspension. The single-cell suspension was then treated with ATc (100 ng ml$^{-1}$ final concentration) to initiate target pre-depletion. To generate 1 d pre-depletion culture, the single-cell suspension was diluted back to $OD_{600}$ = 0.5 in a total volume of 40 ml 7H9. The remaining single-cell suspension was used to generate a 5 d pre-depletion culture, with a starting $OD_{600}$ = 0.1 (40 ml; 100 ng ml$^{-1}$ ATc). After 4 d, the 5 d pre-depletion start culture was further diluted back to a starting $OD_{600}$ = 0.05 (40 ml; in 100 ng ml$^{-1}$ ATc) and incubated for a further 5 d to generate the 10 d pre-depletion culture.

To initiate the chemical-genetic screen, we first collected 10 $OD_{600}$ units of bacteria (~3 × 10$^9$ bacteria; ~30,000× coverage of the CRISPRi library) from the 1, 5 or 10 d CRISPRi library pre-depletion cultures as input controls. Triplicate cultures were then inoculated at $OD_{600}$ = 0.05 in 10 ml 7H9 supplemented with ATc (100 ng ml$^{-1}$), kanamycin (10 µg ml$^{-1}$), and the indicated drug concentration or dimethyl sulfoxide (DMSO) vehicle control (Extended Data Fig. 1). Pooled CRISPRi chemical-genetic screens were performed in vented tissue culture flasks (T-25; Falcon 353109). Cultures were outgrown for 14 d at 37 °C, 5% $CO_2$. ATc was replenished at 100 ng ml$^{-1}$ at day 7. After 14 d outgrowth, $OD_{600}$ values were measured for all cultures to empirically determine the MIC for each drug. Samples from three descending doses of partially inhibitory drug concentrations were processed for genomic DNA extraction, defined as 'High', 'Medium' and 'Low' in Extended Data Fig. 1, as described below. Due to an error during genomic DNA extraction, ethambutol 1,804 nM ('Med') day 1 data reflects two biological replicates, one of which was sequenced twice to produce 3 replicates. Additionally, the 10 d sample was lost for the 221 nM ('Med') streptomycin screen.

**Genomic DNA extraction and library preparation for Illumina sequencing.** Genomic DNA was isolated from bacterial pellets using the CTAB-lysozyme method as previously described[15]. Briefly, Mtb pellets (5–30 $OD_{600}$ units) were resuspended in 1 ml PBS + 0.05% Tween-80. Cell suspensions were centrifuged for 5 min at 4,000 × g and the supernatant was removed. Pellets were resuspended in 800 µl TE buffer (10 mM Tris pH 8.0, 1 mM EDTA) + 15 mg ml$^{-1}$ lysozyme (Alfa Aesar J60701-06) and incubated at 37 °C for 16 h. Next, 70 µl 10% SDS (Promega V6551) and 5 µl proteinase K (20 mg ml$^{-1}$, Thermo Fisher 25530049) were added and samples were incubated at 65 °C for 30 min. Subsequently, 100 µl 5 M NaCl and 80 µl 10% CTAB (Sigma Aldrich H5882) were added, and samples were incubated for an additional 30 min at 65 °C. Finally, 750 µl ice-cold chloroform was added and samples were mixed. After centrifugation at 16,100 × g and extraction of the aqueous phase, samples were removed from the biosafety level 3 facility. Samples were then treated with 25 µg RNase A (Bio Basic RB0474) for 30 min at 37 °C, followed by extraction with phenol:chloroform:isoamyl alcohol (pH 8.0, 25:24:1, Thermo Fisher BP1752I-400), then chloroform. Genomic DNA was precipitated from the final aqueous layer (600 µl) with the addition of 10 µl 3 M sodium acetate and 360 µl isopropanol. DNA pellets were spun at 21,300 × g for 30 min at 4 °C and washed 2× with 750 µl 80% ethanol. Pellets were dried and resuspended with elution buffer (Qiagen 19086) before spectrophotometric quantification.

The concentration of isolated genomic DNA was quantified using the DeNovix double-stranded DNA high sensitivity assay (KIT-DSDNA-HIGH-2; DS-11 series spectrophotometer/fluorometer). Next, the sgRNA-encoding region was amplified from 500 ng genomic DNA with 17 cycles of PCR using NEBNext Ultra II Q5 master mix (NEB M0544L) as described in ref. [15]. Each PCR reaction contained a pool of forward primers (0.5 µM final concentration) and a unique indexed reverse primer (0.5 µM)[15]. Forward primers contain a P5 flow cell attachment sequence, a standard Read1 Illumina sequencing primer binding site and custom stagger sequences to guarantee base diversity during Illumina sequencing. Reverse primers contain a P7 flow cell attachment sequence, a standard Read2 Illumina sequencing primer binding site and unique barcodes to allow for sample pooling during deep sequencing.

Following PCR amplification, each ~230 bp amplicon was purified using AMPure XP beads (Beckman–Coulter A63882) using one-sided selection (1.2×). Bead-purified amplicons were further purified on a Pippin HT 2% agarose gel cassette (target range 180–250 bp; Sage Science HTC2010) to remove carry-over primer and genomic DNA. Eluted amplicons were quantified with a Qubit 2.0 fluorometer (Invitrogen), and amplicon size and purity were quality controlled by visualization on an Agilent 2100 bioanalyzer (high sensitivity chip; Agilent Technologies 5067–4626). Next, individual PCR amplicons were multiplexed into 10 nM pools and sequenced on an Illumina sequencer according to the manufacturer's instructions. To increase sequencing diversity, a PhiX spike-in of 2.5–5% was added to the pools (PhiX sequencing control v3; Illumina FC-110-3001). Samples were run on the Illumina NextSeq 500, HiSeq 2500 or NovaSeq 6000 platform (single-read 1 ×85 cycles and 6 × i7 index cycles).

**Antibacterial activity measurements.** All compounds were dissolved in DMSO (VWR V0231) and dispensed using an HP D300e digital dispenser in a 384-well plate format. DMSO did not exceed 1% of the final culture volume and was maintained at the same concentration across all samples. CRISPRi strains were growth-synchronized and pre-depleted in the presence of ATc (100 ng ml$^{-1}$) for 5 d before assay for MIC analysis. Cultures were then back-diluted to a starting $OD_{580}$ of 0.05 and 50 µl cell suspension was plated in technical triplicate in wells containing the test compound and fresh ATc (100 ng ml$^{-1}$). For checkerboard assays and MIC assays (non-CRISPRi), cultures were growth-synchronized to late log-phase and back-diluted to an $OD_{600}$ of 0.025 before plating (no ATc). Plates were incubated standing at 37 °C with 5% $CO_2$. $OD_{600}$ was evaluated using a Tecan Spark plate reader at 10–14 d post-plating and percent growth was calculated relative to the DMSO vehicle control for each strain. $IC_{50}$ measurements were calculated using a nonlinear fit in GraphPad Prism. For all MIC curves, data represent the mean ± s.e.m. for technical triplicates. Data are representative of at least two independent experiments.

To quantify growth phenotypes on 7H10 agar, 10-fold serial dilutions of Mtb cultures ($OD_{600} = 0.6$) were spotted on 7H10 agar containing drugs at the indicated concentrations and/or ATc at $100\,ng\,ml^{-1}$. Plates were incubated at 37 °C and imaged after 2 weeks.

**Bone marrow-derived macrophage infections.** Bone marrow-derived macrophages (BMDMs) were differentiated from four wild-type, female C57BL/6NTAC mice (Taconic Farms, 6–8 weeks old). All animal work was performed in accordance with the Guide for the Care and Use of Laboratory Animals of the National Institutes of Health, with approval from the Institutional Animal Care and Use Committee of Rockefeller University. Femurs and tibias were collected, and crushed with a sterile mortar and pestle as described[80]. After red blood cell lysis and counter-selection of resident macrophages, bone marrow cells were incubated in the presence of DMEM ($4.5\,g\,l^{-1}$ glucose + L-glutamine + sodium pyruvate; Corning 10–013-CV) + 10% FBS (Sigma Aldrich F4135, Lot no. 17B189) + 15% conditioned L929 cell medium (LCM) and differentiated for 7 d at 37 °C, 5% $CO_2$. Macrophages were then lifted using gentle cell scraping. For infection assays, BMDMs were seeded in 96-well plates at 75,000 cells per well 2 d before infection. At 16 h before infection, fresh DMEM + 10% FBS + 10% LCM was added to cells, with or without IFN-γ ($20\,ng\,ml^{-1}$; Gemini Biosciences 300-311P). Mtb cultures were synchronized to late log-phase ($OD_{600}$ 0.6–0.8). For infections with CRISPRi strains, cultures were pre-depleted with $100\,ng\,ml^{-1}$ ATc for 24 h before infection. Mtb pellets were washed with PBS (Thermo Fisher 14190144) + 0.05% Tyloxapol (Sigma Aldrich T0307) and single-cell suspensions were generated by collecting the suspended cells after gentle centrifugation ($150 \times g$ for 12 min). Cell culture medium was removed from the macrophages and replaced with Mtb-containing medium at a multiplicity of infection of 1:1. After 4 h of infection at 37 °C, the medium was removed and cells were washed 2× with PBS. Wells were replenished with fresh media with or without drug. DMSO was normalized to 0.2%. For CRISPRi infections, doxycycline (Sigma Aldrich D9891) was added at a concentration of $250\,ng\,ml^{-1}$ to maintain target knockdown. For all infection assays, medium was replaced with fresh drug at day 3. At each indicated timepoint, after two PBS washes, cells were lysed with $100\,\mu l$ Triton X-100 in water (Sigma Aldrich X100). Lysates were titrated in PBS + 0.05% Tween-80 and plated on 7H10 containing activated charcoal ($4\,g\,l^{-1}$) to absorb any residual ATc or antibiotic. Colony-forming units were enumerated after 21–28 d of outgrowth. Data for BMDM experiments represent the mean ± s.e.m. for technical triplicates. Results are representative of at least two independent experiments.

**Selection of drug-resistant Mtb isolates.** For the selection of linezolid-resistant H37Rv Mtb mutants, two independent cultures were started at an $OD_{600}$ of 0.001. After 1 week of outgrowth, cultures were pelleted and roughly $3 \times 10^9$ c.f.u. were plated on 7H10 + 11.9 μM linezolid. Plates were incubated for 24 d. Colonies were picked and grown in 7H9 + 11.9 μM linezolid. Genomic DNA was isolated as described above. Sanger sequencing was performed on purified PCR amplicons of *rrl* (23S rRNA) and *rplC* using the primers listed in Supplementary Table 2. Genomic DNA was also submitted for whole genome sequencing to confirm the absence of other mutations that may contribute to linezolid resistance (BGI, llumina HiSeq X Ten platform).

For selection of resistant isolates for the lineage 1.2.1 strain, 30 independent 20 ml cultures were started at an $OD_{600}$ of 0.001. After growth to log-phase ($OD_{600}$ 0.5–0.6), cultures were pelleted and plated on 7H10 + antibiotic ($CLR = 10\,\mu g\,ml^{-1}$, $RIF = 0.5\,\mu g\,ml^{-1}$). After 28–35 d, colonies were enumerated. CLR-resistant colonies occurred at a frequency of $1.7 \times 10^{-8}$ and were picked and grown in 7H9 + CLR ($4\,\mu g\,ml^{-1}$). Samples were heat-lysed, and *whiB7* and *rrl* were PCR-amplified and sequenced using the primers listed in Supplementary Table 2. Sanger sequencing revealed a parental *whiB7* locus (Gly64delG) in all ($n = 101$) clarithromycin-resistant isolates. Instead, clarithromycin resistance was conferred by a variety of base substitutions in the 23S rRNA (Extended Data Fig. 9h). Select samples were cultured further and purified genomic DNA was submitted for whole genome sequencing.

**Total RNA extraction and RT-qPCR.** Total RNA extraction was performed as previously described[15]. Briefly, 2 $OD_{600}$ units of bacteria were added to an equivalent volume of GTC buffer (5 M guanidinium thiocyanate, 0.5% sodium N-lauroylsarcosine, 25 mM trisodium citrate dihydrate and 0.1 M 2-mercaptoethanol), pelleted by centrifugation, resuspended in 1 ml TRIzol (Thermo Fisher 15596026) and lysed by zirconium bead beating (MP Biomedicals 116911050). Chloroform (0.2 ml) was added to each sample and samples were frozen at −80 °C. After thawing, samples were centrifuged to separate phases and the aqueous phase was purified by Direct-zol RNA miniprep (Zymo Research R2052). Residual genomic DNA was removed by TURBO DNase treatment (Invitrogen Ambion AM2238). After RNA cleanup and concentration (Zymo Research R1017), 3 μg RNA per sample was reverse transcribed into complementary DNA (cDNA) with random hexamers (Thermo Fisher 18-091-050) following the manufacturer's instructions. RNA was removed by alkaline hydrolysis and cDNA was purified with PCR cleanup columns (Qiagen 28115). Next, knockdown of the targets was quantified by SYBR green dye-based quantitative real-time PCR (Applied Biosystems 4309155) on a Quantstudio System 5

(Thermo Fisher A28140) using gene-specific qPCR primers (5 μM), normalized to *sigA* (*rv2703*) and quantified by the ΔΔCt algorithm. All gene-specific qPCR primers were designed using the PrimerQuest tool from IDT (https://www.idtdna.com/PrimerQuest/Home/Index) and then validated for efficiency and linear range of amplification using standard qPCR approaches. Specificity was confirmed for each validated qPCR primer pair through melting curve analysis.

**Mouse infection and drug treatment.** Female BALB/c mice (Charles Rivers Laboratory, 7–8 weeks old) were infected with 100–200 c.f.u. of Mtb using a whole-body inhalation exposure system (Glas-Col). After 10 d (Fig. 6d,e) or 14 d (Extended Data Fig. 9e,f), animals were randomly assigned to study groups and chemotherapy was initiated. Depending on the experiment, either 5 or 6 mice were allocated to each treatment condition (as indicated in the figure legend). Clarithromycin and rifampicin were stirred in 0.5% carboxymethylcellulose (CMC)/0.5% Tween-80 for resuspension. Isoniazid and azithromycin were resuspended in water. Liquid drug formulations were administered once daily by oral gavage for 14 consecutive days. After 13 d of drug treatment, blood samples of the mice were taken 1 h and 24 h post dosing (Extended Data Fig. 9c,d). At designated timepoints after starting chemotherapy mice were euthanized, lungs and spleens were aseptically removed, homogenized in 1 ml PBS + 0.05% Tween-80 and plated on Middlebrook 7H11 agar supplemented with 10% OADC. Colonies were counted after 4–6 weeks of incubation at 37 °C. Mice were housed in groups of 5 in individually ventilated cages inside a certified ABSL-3 facility and had access to water and food ad libitum for the duration of the study. All experiments involving animals were approved by the Institutional Animal Care and Use Committee of the Center for Discovery and Innovation.

**Drug quantitation in plasma by high-pressure liquid chromatography coupled to tandem mass spectrometry (LC–MS/MS).** Neat $1\,mg\,ml^{-1}$ DMSO stocks for rifampicin, azithromycin (AZM) and clarithromycin were serial-diluted in 50/50 (acetonitrile/water) to create neat spiking stocks. Standards and quality controls were created by adding 10 μl of spiking stock to 90 μl of drug-free plasma. Control, standard, quality control or study sample (10 μl) were added to 100 μl 50/50 (acetonitrile/methanol) protein precipitation solvent containing the stable labelled internal standards RIF-d8 (Toronto Research Chemicals R508003), AZM-d5 (Toronto Research Chemicals A927004) and CLR-13C-d3 (Cayman Chemical 26678) at $10\,ng\,ml^{-1}$. Extracts were vortexed for 5 min and centrifuged at $1,500 \times g$ for 5 min. The supernatant of RIF (100 μl) containing samples was combined with 5 μl $75\,mg\,ml^{-1}$ ascorbic acid to stabilize RIF. The mixture (100 μl) was combined with 100 μl Milli-Q water before HPLC–MS/MS analysis. CD-1 mouse control plasma ($K_2$EDTA) was sourced from Bioreclamation. RIF, AZM and CLR were sourced from Sigma Aldrich.

LC–MS/MS analysis was performed on a Sciex Applied Biosystems Qtrap 6500+ triple-quadrupole mass spectrometer coupled to a Shimadzu Nexera X2 UHPLC system to quantify each drug in plasma. Chromatography was performed on an Agilent SB-C8 (2.1 ×30 mm; particle size, 3.5 μm) using a reverse phase gradient. Milli-Q deionized water with 0.1% formic acid was used for the aqueous mobile phase and 0.1% formic acid in acetonitrile for the organic mobile phase. Multiple-reaction monitoring of precursor/product transitions in electrospray positive-ionization mode was used to quantify the analytes. Sample analysis was accepted if the concentrations of the quality control samples were within 20% of the nominal concentration. The compounds were ionized using electrospray positive-ionization mode and monitored using masses for RIF (823.50/791.60), AZM (749.38/591.30), CLR (748.38/158.20), RIF-d8 (831.50/799.60), AZM-d5 (754.37/596.30) and CLR-d4 (752.33/162.10). Data processing was performed using Analyst software (version 1.6.2; Applied Biosystems Sciex).

**Reporting summary.** Further information on research design is available in the Nature Research Reporting Summary linked to this article.

## Data availability

Raw sequencing data are deposited to the NCBI Short Read Archive under project number PRJNA738381. All screen results are available in Source Data Fig. 1 and at pebble.rockefeller.edu. Source data are provided with this paper.

## Code availability

The custom code used in this study is available at https://github.com/rock-lab/CGI_nature_micro_2022.

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

## Acknowledgements

We thank members of the Rock laboratory, S. Ehrt, J. Pritchard, A. Gouzy and S. Grover for comments on the manuscript and/or helpful discussions; C. Trujillo, J. Wallach, S. Lavalette-Levi and S. Ehrt for technical assistance, J. Xiang and D. Xu of the Weill Cornell Genomics Core for sequencing; the animal technical and analytical teams of the Center for Discovery and Innovation for animal work; R. Froom and V. LaSalle for scientific illustration; and S. Schrader and C. Barry for sharing bacterial strains. This work was supported by a Harvey L. Karp Postdoctoral Fellowship (S.L.), the Potts Memorial Foundation (M.A.D. and S.L.), the Bill and Melinda Gates Foundation (INV-010616 and INV-004761, D.S.), the Department of Defense (PR192421, J.M.R.), the Robertson Therapeutic Development Fund (J.M.R.), an NIH Shared Instrumentation Grant (S10-OD023524, V.A.D.) and an NIH/NIAID New Innovator Award (1DP2AI144850-01, J.M.R.).

## Author contributions

S.L., N.C.P., N.R., D.S. and J.M.R. conceptualized the study; S.L., N.C.P., J.S.C., Z.A.A., M.A.D. and J.M.R. developed the methodology; S.L., N.C.P., N.R., M.D.Z., B.B., C.A.E., D.F.S. and M.G. conducted the investigation; S.L. and N.C.P. conducted validation; J.S.C., M.A.D., Z.A.A. and K.A.E. conducted software and formal analysis; J.S.C., M.A.D., Z.A.A., K.A.E. and J.M.R. curated the data; S.L., N.C.P. and J.M.R. wrote the original draft of the paper; S.L., N.C.P., J.S.C., M.A.D., K.A.E., B.B., M.G., V.A.D., D.S. and J.M.R. reviewed and edited the paper; V.A.D., D.S. and J.M.R. acquired funding; M.D.Z., M.G. and V.D. acquired resources; J.M.R supervised the work.

## Competing interests

All authors declare no competing interests.

## Additional information

**Extended data** is available for this paper at https://doi.org/10.1038/s41564-022-01130-y.

**Correspondence and requests for materials** should be addressed to Jeremy M. Rock.

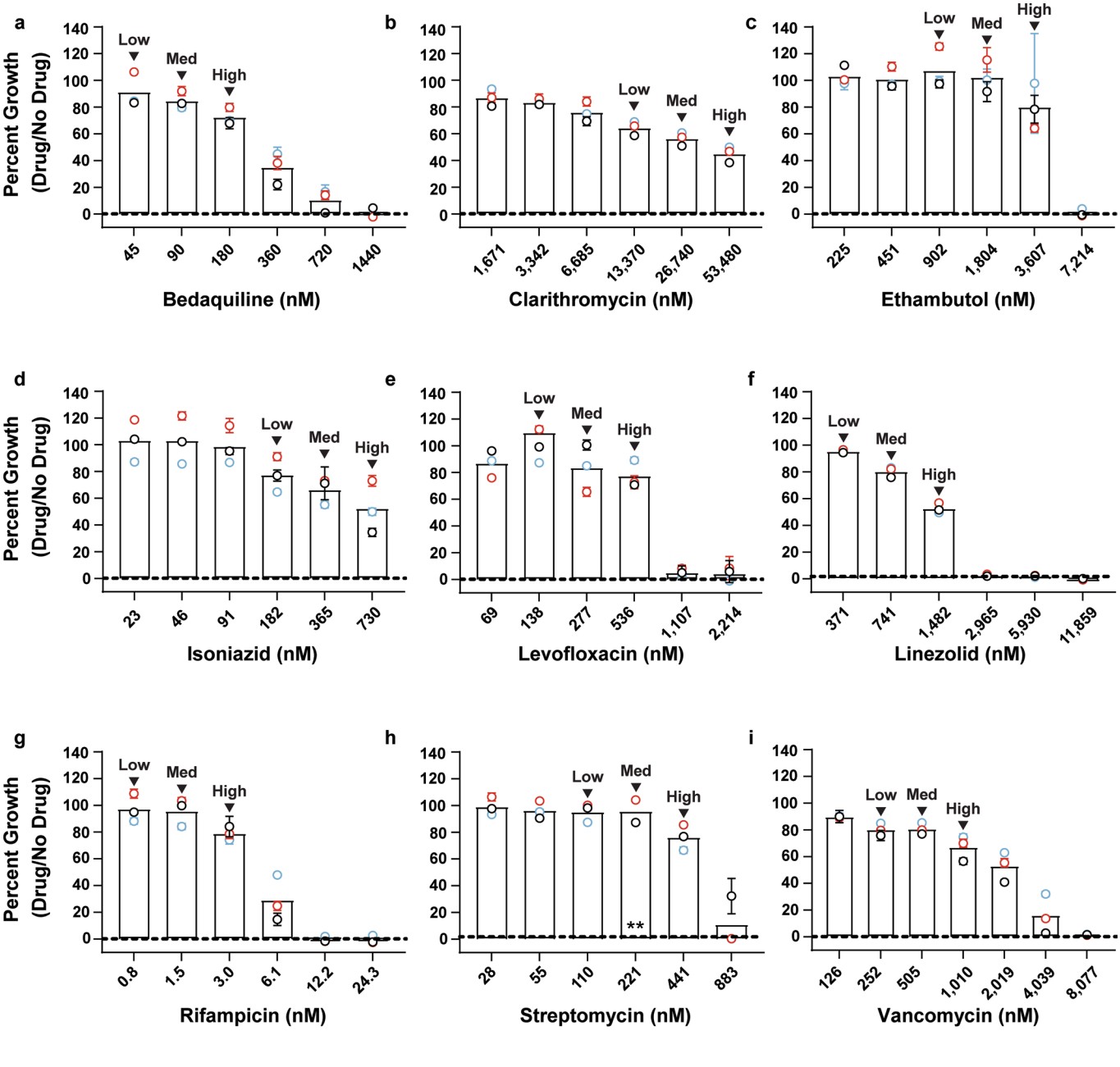

**Extended Data Fig. 1 | Growth of the Mtb CRISPRi library during drug selection. a-i**, Normalized growth (mean ± SEM, n = 3 biological replicates) of the Mtb CRISPRi library in the drug screens. Samples harvested for sgRNA deep sequencing are marked as "High", "Med", and "Low", denoting the three descending doses of partially inhibitory drug concentrations analyzed in these screens. **: 10-day sample was lost for the 221 nM ("Med") streptomycin screen. One biological replicate was lost for the 1,804 nM ("Med") EMB 1-day screen; to obtain three replicate samples, one biological replicate was prepared in technical duplicate.

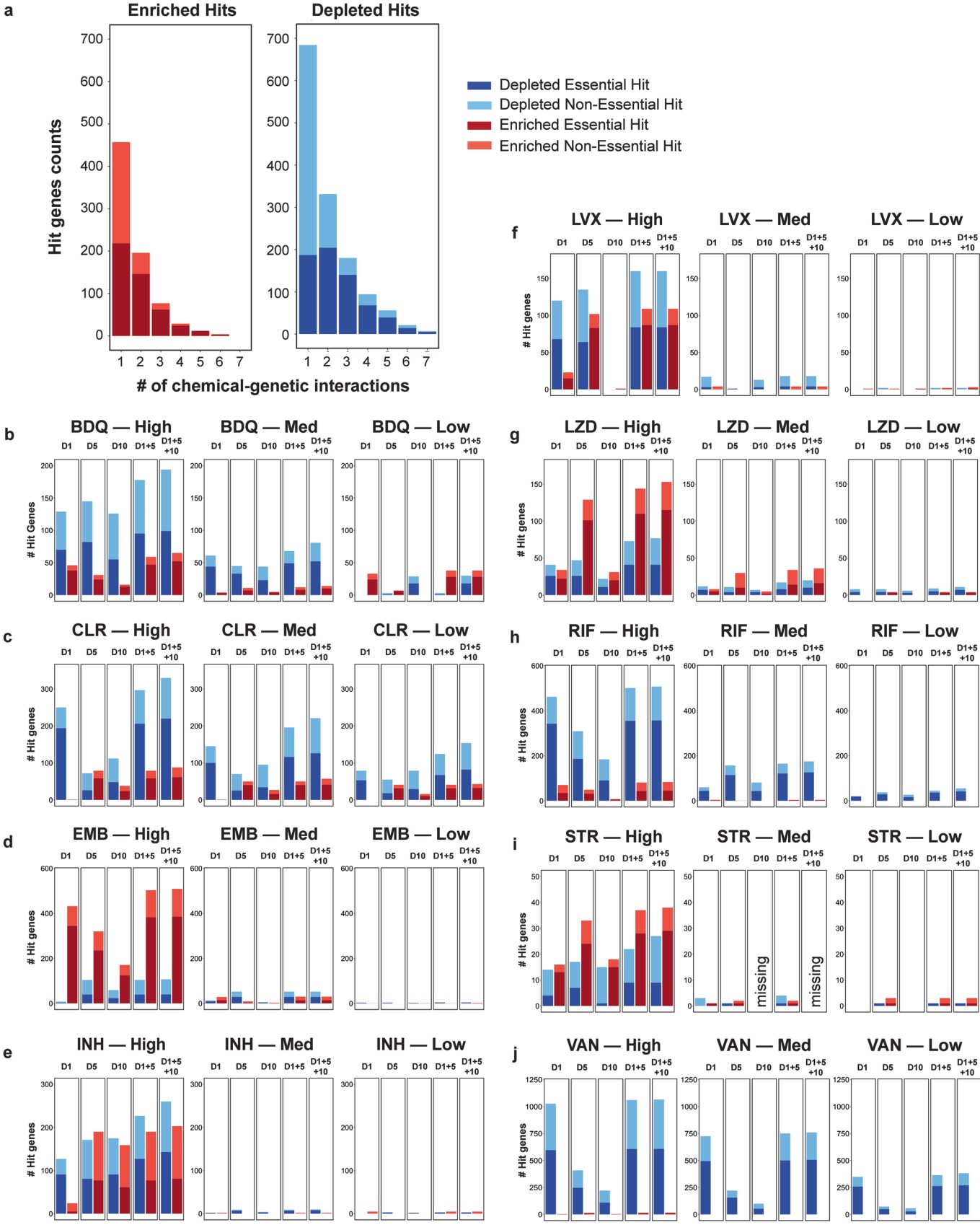

**Extended Data Fig. 2 | See next page for caption.**

**Extended Data Fig. 2 | Summary of hits from chemical-genetic screens. a**, Histogram depicting the number of unique chemical-genetic interactions for enriching and depleting hits. Hit genes were defined as the union of 1 and 5-day CRISPRi library pre-depletion results. **b-j**, Bar graphs showing the number of hit genes identified across all drugs. Gene essentiality calls were defined by CRISPRi[15]. D1, D5, and D10 indicate the number of days the CRISPRi library was treated with ATc prior to drug exposure; D1 + 5 = hit genes defined as the union of 1 and 5-day CRISPRi library pre-depletion results; D1 + 5 + 10: hit genes defined as the union of 1, 5 and 10 day CRISPRi library pre-depletion results. The 10-day sample was lost for the "Med" streptomycin screen and thus the D10 containing results for "STR–Med" are labelled "missing". BDQ = bedaquiline; CLR = clarithromycin; EMB = ethambutol; INH = isoniazid; LVX = levofloxacin; LZD = linezolid; RIF = rifampicin, STR = streptomycin; VAN = vancomycin.

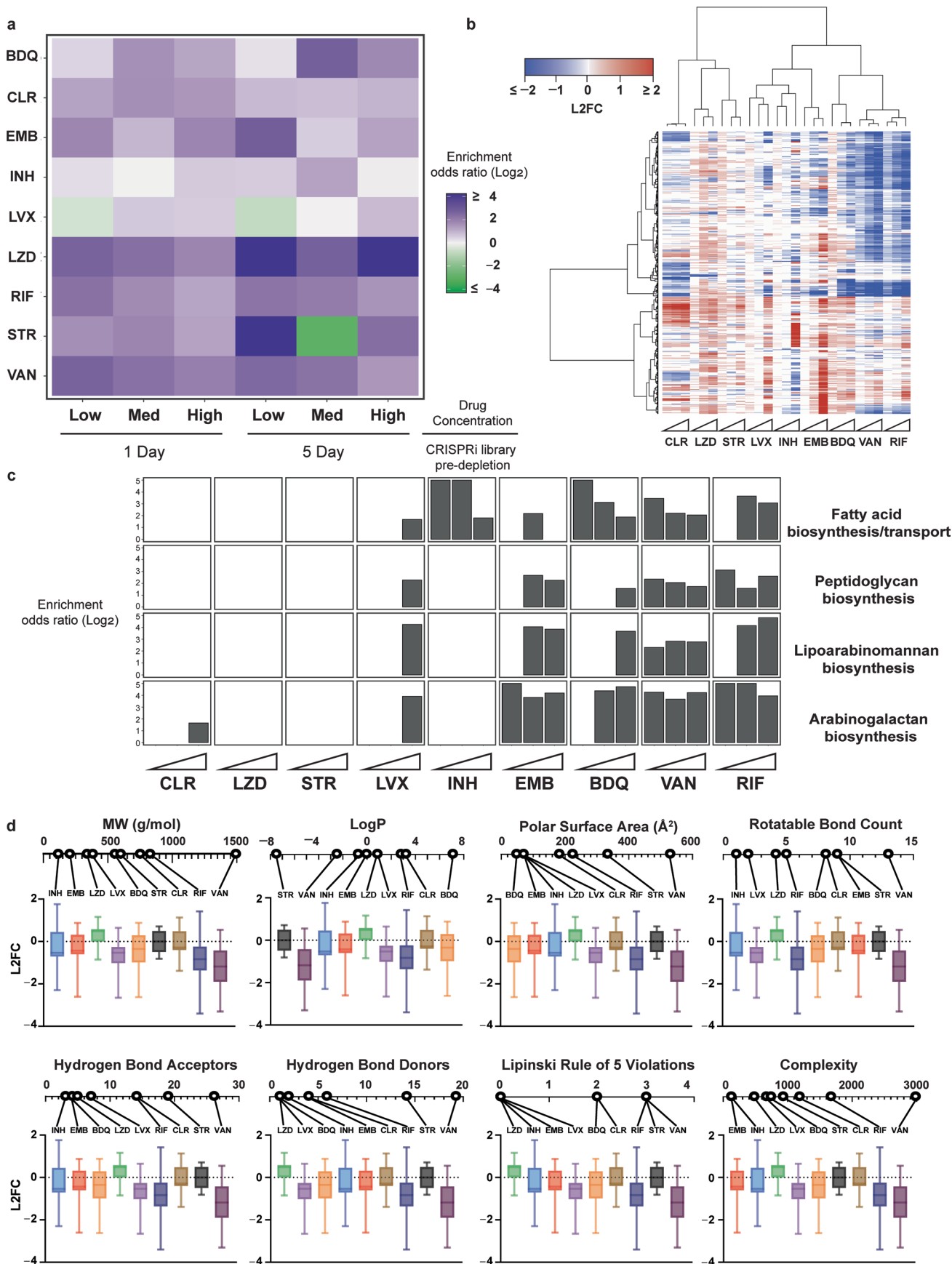

**Extended Data Fig. 3 | See next page for caption.**

**Extended Data Fig. 3 | Clustering & enrichment analysis of chemical-genetic profiles. a**, Heatmap of odds-ratios showing enrichment of essential gene targeting sgRNAs as hits in the chemical-genetic screen. A Fisher's exact test was used to evaluate enrichment of essential gene targeting sgRNAs relative to non-essential gene targeting sgRNAs amongst hit genes (FDR < 0.01, |L2FC| > 1) in the chemical genetic screen. BDQ = bedaquiline; CLR = clarithromycin; EMB = ethambutol; INH = isoniazid; LVX = levofloxacin; LZD = linezolid; RIF = rifampicin, STR = streptomycin; VAN = vancomycin. **b**, Heatmap showing clustered chemical-genetic profiles from the 5-day CRISPRi library pre-depletion screen. Genes are clustered along the vertical axis; for simplicity, only genes that hit in at least two drug conditions are shown (n = 676 genes). Ascending drug concentrations ("Low", "Med", "High" indicated by white triangles) are clustered along the horizontal axis. The median L2FC for each gene following drug selection (relative to vehicle control) is indicated on the color scale. If a gene was not a significant hit (FDR > 0.01), the L2FC value was plotted as 0 for the corresponding condition. **c**, Bar plots of the enriched (P < 0.05) KEGG categories for hit genes for the indicated drugs. KEGG annotations were manually updated to include the mycolic acid-arabinogalactan-peptidoglycan (mAGP) complex-associated genes described in[20,21]. **d**, Correlation of mAGP signature and drug physiochemical properties. For each drug, the L2FC distribution ("High" concentration, 5-day CRISPRi library pre-depletion) is shown for a select group of 78 genes involved in mAGP assembly and regulation as described in[20,21]. L2FC values (mean ± SEM, n = 3 biological replicates) for the 78 genes under each drug treatment are shown. For each condition, the box indicates the lower quartile, median, and upper quartile and whiskers represent the minimum and maximum L2FC values.

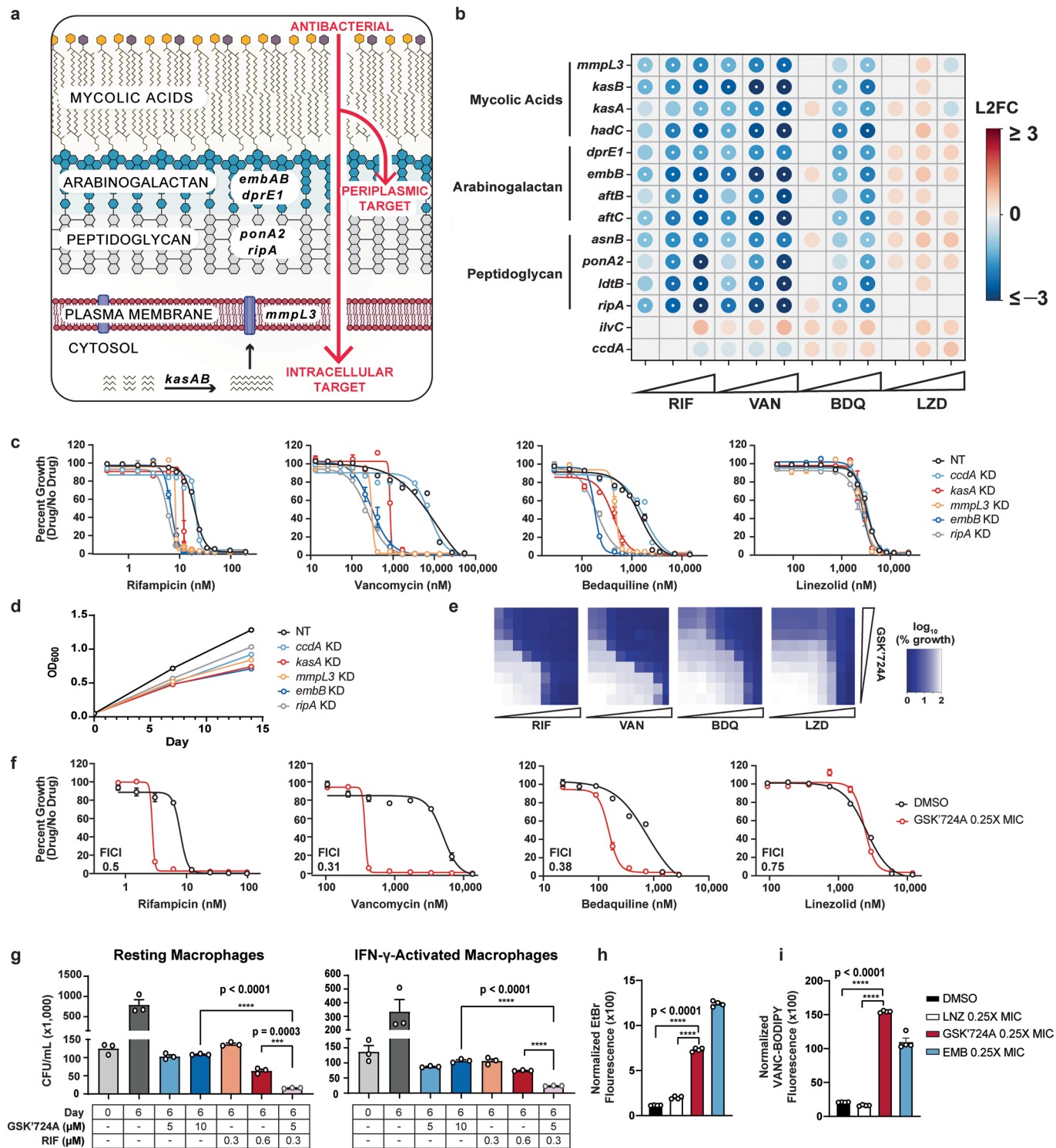

**Extended Data Fig. 4 | See next page for caption.**

**Extended Data Fig. 4 | The Mtb envelope mediates intrinsic resistance to a subset of drugs. a**, Diagram of the mycobacterial mAGP complex. Select genes involved in mycolic acid synthesis and transport (*kasAB, mmpL3*), arabinogalactan biosynthesis (*embAB, dprE1*), and peptidoglycan remodeling (*ponA2, ripA*) are highlighted. **b**, Heatmap of chemical-genetic hit genes from the 5-day CRISPRi library pre-depletion screen. The color of each circle represents the gene-level L2FC. A white dot represents an FDR < 0.01 and a|L2FC| > 1. *ilvC* and *ccdA* are included as non-hit controls. **c,d**, Single strain validation of mAGP-associated hits. Dose-response curves (**c**, mean ± SEM, n = 3 biological replicates) for the indicated hypomorphic CRISPRi strains. Growth curves (**d**, mean ± SEM, n = 3 biological replicates) are derived from the vehicle control samples. NT = non-targeting; KD = knockdown. **e,f**, KasA inhibitor (GSK'724 A) checkerboard assays to quantify drug-drug interactions. Dose-response curves (**f**, mean ± SEM, n = 3 biological replicates) are shown for each drug in the absence (DMSO) or presence of 0.25X $MIC_{80}$ GSK'724 A. The fractional inhibitory concentration index (FICI) values listed represent the lowest value obtained from each checkerboard assay. **g**, GSK'724 A synergy with rifampicin in resting and IFN-γ-activated murine bone marrow derived macrophages (mean CFU/ml ± SEM, n = 3 biological replicates). Results from an unpaired, two-tailed t-test are shown. **h,i**, Ethidium bromide (**h**) and Vancomycin-BODIPY (**i**) uptake (mean ± SEM, n = 4 biological replicates) of H37Rv pre-treated for two days with DMSO or subinhibitory linezolid, GSK'724 A, or ethambutol. Results from an unpaired, two-tailed t-test are shown.

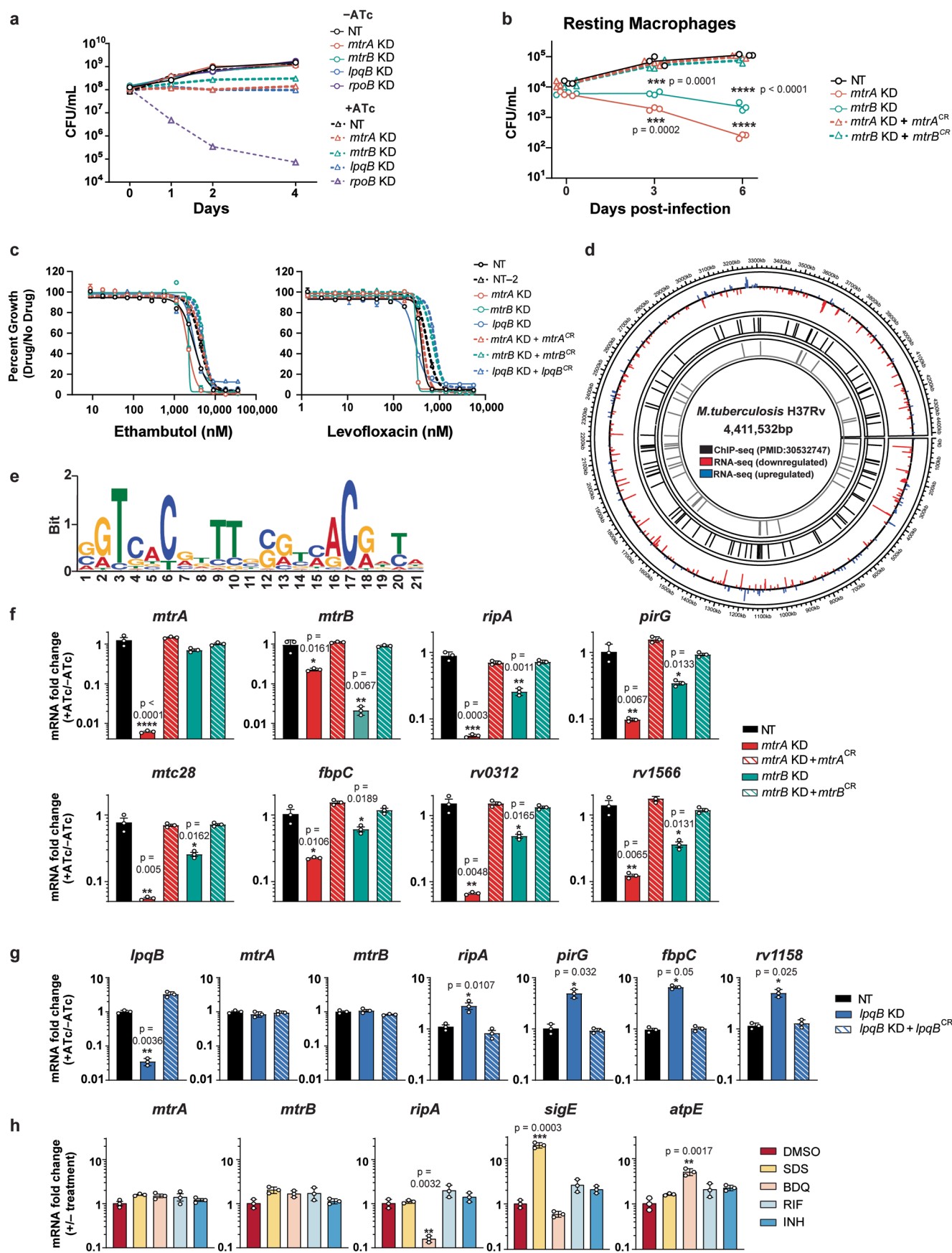

**Extended Data Fig. 5 | See next page for caption.**

**Extended Data Fig. 5 | The MtrA response regulator is critical for multidrug intrinsic resistance. a**, Time-kill curves (mean ± SEM, n = 3 biological replicates) for the indicated CRISPRi strains. NT = non-targeting; KD = knockdown; CR = CRISPRi-resistant. **b**, Growth (mean ± SEM, n = 3 biological replicates) of the indicated CRISPRi strains in resting murine bone marrow derived macrophages. Significance was determined by two-way ANOVA and adjusted for multiple comparisons. ***, $p < 0.001$; ****, $p < 0.0001$. **c**, Dose-response curves (mean ± SEM, n = 3 biological replicates) for the indicated strains. **d**, Circos plot depicting overlapping genes identified by RNA-seq (Fig. 2f) and MtrA ChIP-seq[27]. Outer track: the H37Rv genome by nucleotide position; middle track: lines mark genes with significant L2FC values (padj < 0.05) upon *mtrA* knockdown (blue = positive L2FC; red = negative L2FC); inner tracks: black lines mark genes defined as interacting with MtrA by ChIP-seq[27], and grey lines highlight genes which display both a significant expression change (padj < 0.05; |L2FC| > 1) by *mtrA* RNAseq and are found to interact with MtrA by ChIP-seq. **e**, Identification of an MtrA consensus binding motif. MEME analysis was performed on the promoter regions of candidate genes found to be downregulated upon *mtrA* silencing and bound by MtrA by ChIP-seq[27] (n = 25 genes). **f,g**, Quantification (mean ± SEM, n = 3 biological replicates) of gene mRNA levels by qRT-PCR. Strains were grown +/− ATc for ~3 generations prior to harvesting RNA. Statistical significance was calculated as p-value with unpaired t-test. *, $p < 0.05$; **, $p < 0.01$; ***, $p < 0.001$. **h**, Quantification (mean ± SEM, n = 3 biological replicates) of gene mRNA levels by qRT-PCR. Wild-type H37Rv was grown in the presence of the indicated stress (RIF/BDQ/INH: 4X $IC_{50}$, SDS: 0.2%, DMSO: 0.5%) for 3 hours prior to harvesting RNA. Statistical significance was calculated as p-value with unpaired T-test. ***, $p < 0.001$.

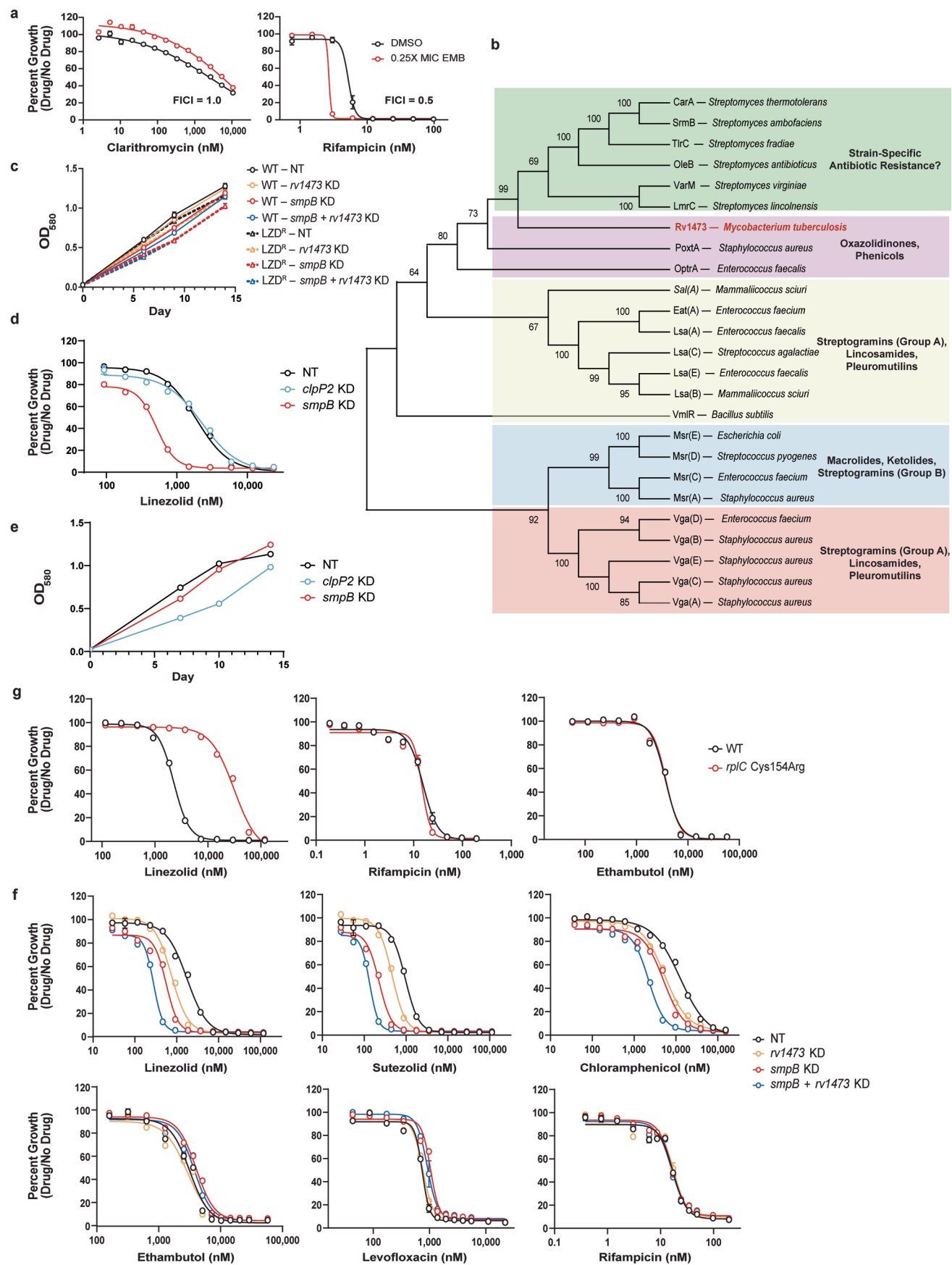

**Extended Data Fig. 6 | See next page for caption.**

**Extended Data Fig. 6 | Mtb encodes diverse mechanisms of intrinsic resistance to ribosome-targeting antibiotics. a**, Ethambutol checkerboard assays to quantify drug-drug interactions. Dose-response curves (mean ± SEM, n = 3 biological replicates) are shown for each drug in the absence (DMSO) or presence of 0.25X MIC$_{80}$ of EMB. Fractional inhibitory concentration index (FICI) values listed represent the lowest value obtained from each checkerboard assay. **b**, Phylogenetic tree of antibiotic resistance (ARE) ABC-F proteins from the indicated species. Figure adapted from[44] Bootstrap values (500 replicates) are indicated at each node. **c**, Growth curves for the linezolid-associated hit genes and control strains shown in Fig. 3d. Curves (mean ± SEM, n = 3 biological replicates) are derived from the vehicle control samples of the MIC assay. NT = non-targeting; KD = knockdown. **d**, Dose-response curves (mean ± SEM, n = 3 biological replicates) were measured for CRISPRi knockdown strains targeting *smpB* and *clpP2* in wild-type H37Rv. **e**, Growth curves for the strains shown in panel (**d**). Curves (mean ± SEM, n = 3 biological replicates) are derived from the vehicle control samples of the MIC assay. **f**, Dose-response curves (mean ± SEM, n = 6 biological replicates) of WT H37Rv and an isogenic *rplC*-Cys154Arg mutant. **g**, Dose-response curves (mean ± SEM, n = 3 biological replicates) for the indicated CRISPRi strains.

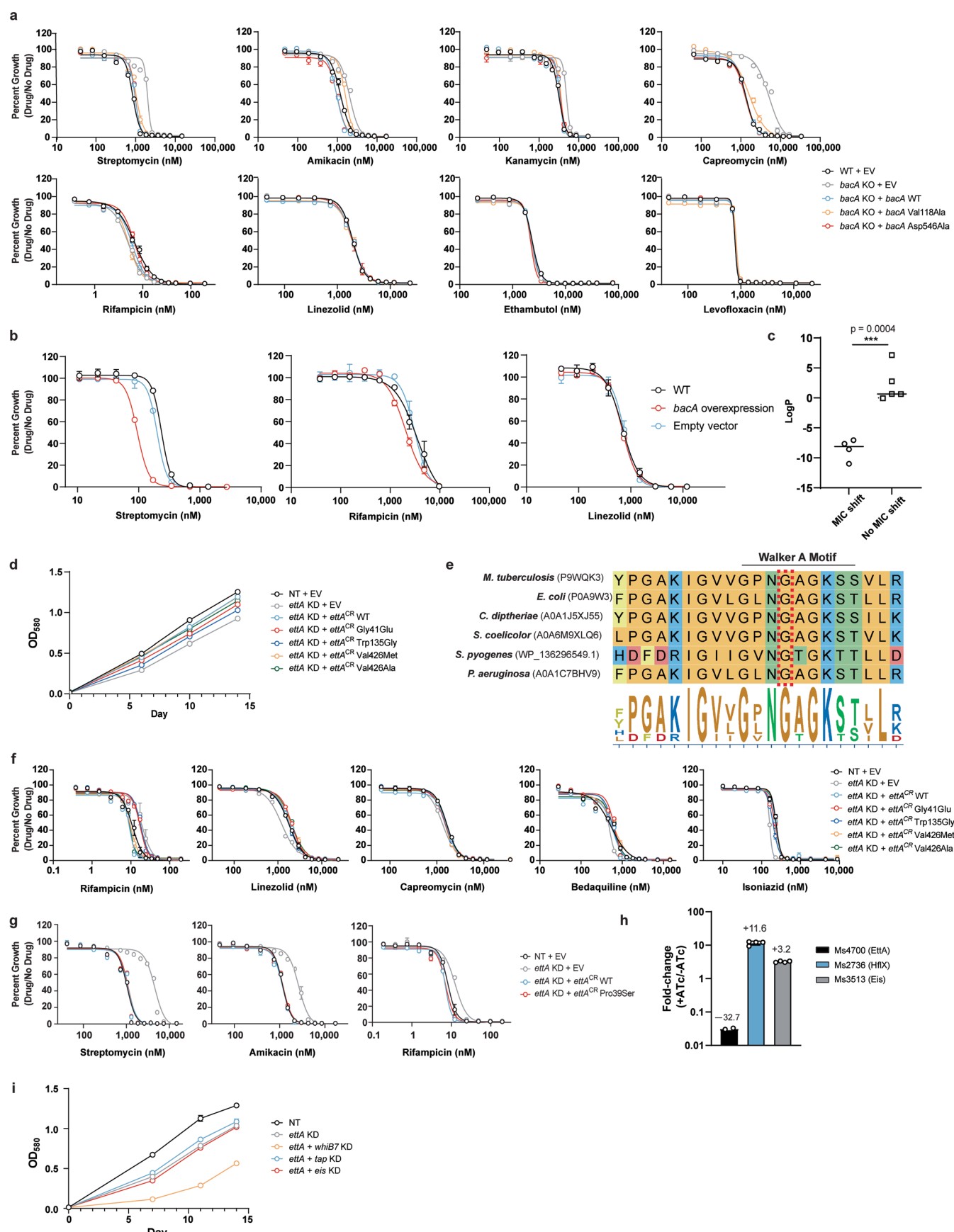

**Extended Data Fig. 7 | See next page for caption.**

**Extended Data Fig. 7 | Loss-of-function mutations in *bacA* and *ettA* confer acquired drug resistance. a**, Dose-response curves (mean ± SEM, n = 3 biological replicates) show that the Val118Ala and Asp546Ala mutations in *bacA* do not confer drug resistance. KO = knockout; EV = empty complementation vector. **b**, Overexpression of Mtb *bacA* confers streptomycin sensitivity in *M. smegmatis*. Dose-response curves (mean ± SEM, n = 3 biological replicates) for the indicated strains. **c**, LogP values for the antibiotics to which *bacA* mutants show an increased MIC (streptomycin, amikacin, capreomycin, kanamycin) or no MIC change (rifampicin, ethambutol, levofloxacin, linezolid). Results from an unpaired t-test are shown: ***, p < 0.001. **d**, Growth curves for the strains shown in Fig. 5c. Curves (mean ± SEM, n = 3 biological replicates) are derived from the vehicle control samples of the MIC assay. KD = knockdown. **e**, Sequence alignment for EttA orthologs from the indicated species for the region surrounding the N-terminal Walker A motif. The Mtb EttA Gly41 residue is boxed. Accession numbers are listed next to each species. **f**, Dose-response curves (mean ± SEM, n = 3 biological replicates) for the strains shown in Fig. 5c. **g**, Dose-response curves (mean ± SEM, n = 3 biological replicates) show that the Pro39Ser mutation in *ettA* does not confer drug resistance. **h**, Quantitative mass spectrometry results following CRISPRi knockdown of *ms4700* (described in[15]). Data represent protein level fold-change (mean ± SEM, n = 4 technical replicates derived from 2 biological replicates). Ms4700 could only be detected in two +ATc replicates and thus the mean ± SEM for duplicates is shown. **i**, Growth curves for the strains shown in Fig. 5e. Curves (mean ± SEM, n = 3 biological replicates) are derived from the vehicle control samples of the MIC assay.

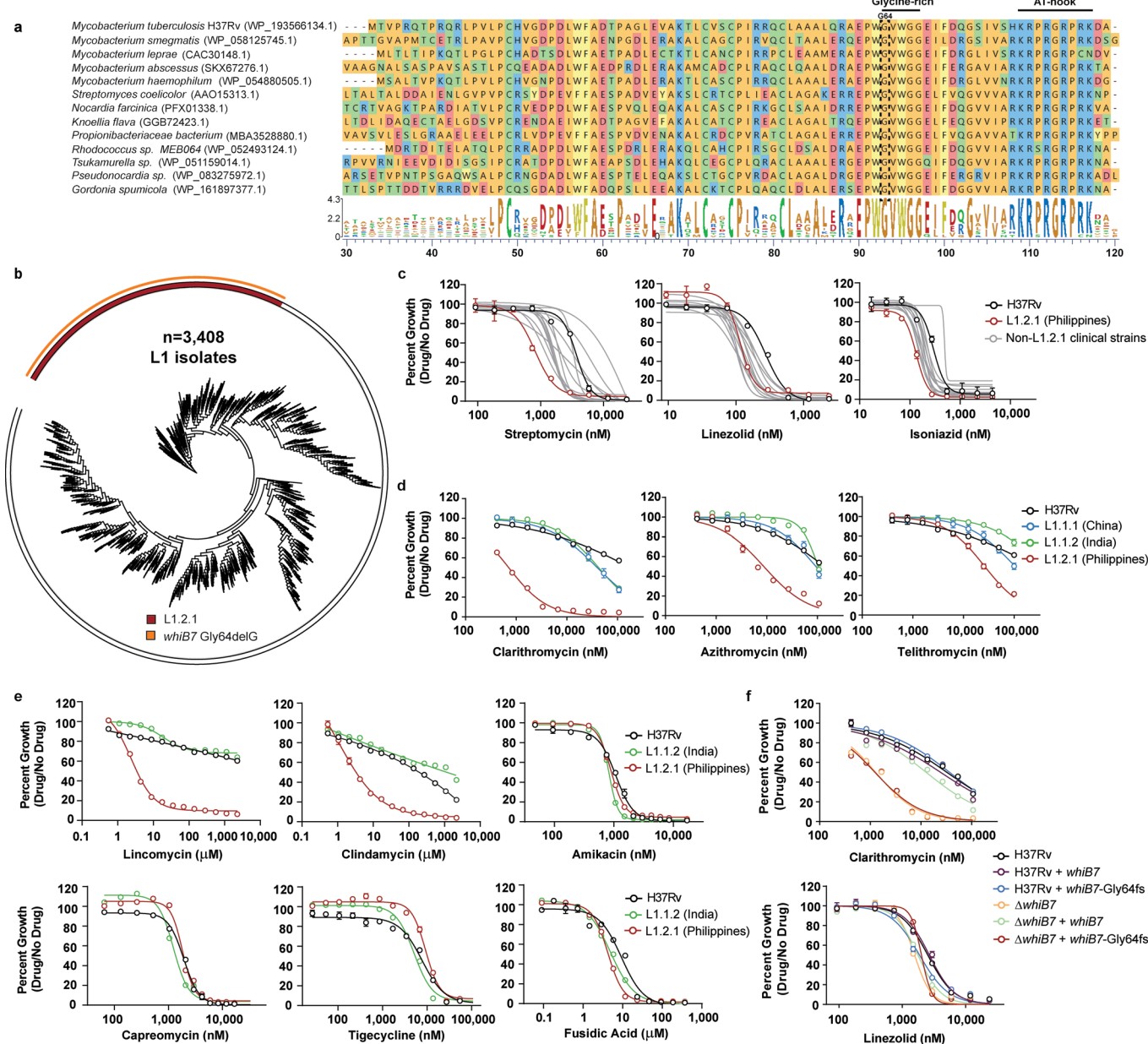

**Extended Data Fig. 8 | L1.2.1 is a *whiB7* loss-of-function mutant and hypersusceptible to macrolides, ketolides, and lincosamides. a**, Alignment of WhiB7 orthologues from representative actinobacteria. Accession numbers are listed next to each species. The conserved glycine-rich motif and DNA binding AT-hook element are highlighted. **b**, Phylogenetic tree of all L1 Mtb clinical isolates (n = 3,408) in our genome database (Source Data Fig. 4). L1.2.1 and the *whiB7* Gly64delG mutation are highlighted. **c-e**, Dose-response curves (mean ± SEM, n = 3 biological replicates) for a reference set of Mtb clinical strains[66]. **f**, H37Rv or a Δ*whiB7* H37Rv strain were transformed with an integrating plasmid to express either the H37Rv *whiB7* allele or the L1.2.1 *whiB7* Gly64delG allele *in trans* from its native promoter. Dose-response curves (mean ± SEM, n = 3 biological replicates) for the resulting strains are shown.

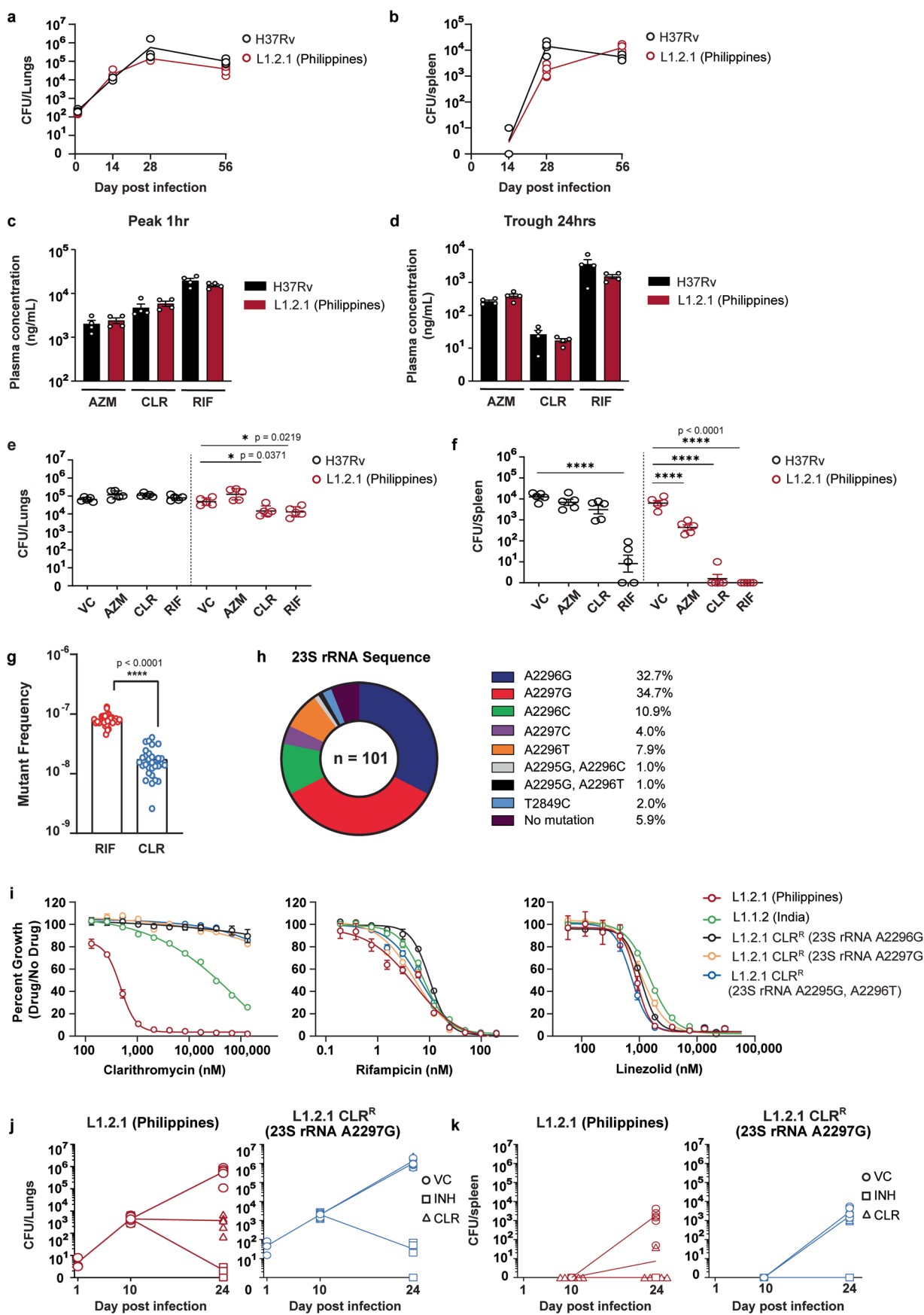

**Extended Data Fig. 9 | See next page for caption.**

**Extended Data Fig. 9 | L1.2.1 is susceptible to clarithromycin *in vivo*. a,b,** Growth of H37Rv and L1.2.1 *in vivo*. BALB/c mice were infected with approximately 100-200 c.f.u. by aerosol. Mtb burden (mean c.f.u. ± SEM) in lungs (**a**) and spleens (**b**) was determined at the indicated timepoints. Data represent 3 mice at D0 (lung only), 5 mice at D14 and D28, and 6 mice at D56. **c,d,** Plasma drug concentrations (mean ± SD for 4 mice) after 2 weeks of drug therapy. Blood was collected at 1 h (**c**) and 24 h (**d**) post-dose after 13 daily doses from vehicle control (VC) and treatment groups further described in (**e,f**). Drug concentrations were measured in plasma using high pressure liquid chromatography coupled to tandem mass spectrometry. AZM = azithromycin; CLR = clarithromycin; RIF = rifampicin. **e,f,** Lung (**e**) and spleen (**f**) Mtb burden (mean c.f.u. ± SEM, n = 5 mice) in BALB/c mice after azithromycin (200 mg/kg), clarithromycin (200 mg/kg), or rifampicin (25 mg/kg) treatment. Mice were infected with approximately 100-200 c.f.u. of aerosolized Mtb. After two weeks to allow the acute infection to establish, chemotherapy was initiated. Following two weeks of drug therapy, Mtb bacterial load was determined. Statistical significance was assessed by one-way ANOVA followed by Tukey's post-hoc test. *, p < 0.05; ****, p < 0.0001. **g,** Rifampicin and clarithromycin mutation frequency analysis with the L1.2.1 strain. Results from an unpaired, two-tailed student's t-test are shown **h,** Distribution of 23 S rRNA mutations from *in vitro*-selected, clarithromycin-resistant L1.2.1 isolates from (**g**). **i,** Dose-response curves (mean ± SEM, n = 3 biological replicates) of representative CLR-resistant (CLR^R) L1.2.1 isolates. **j,k,** Lung (**j**) and spleen (**k**) Mtb burden (mean c.f.u. ± SEM) in BALB/c mice after isoniazid (INH; 25 mg/kg) or clarithromycin (200 mg/kg) treatment. Mice were infected with approximately 100-200 c.f.u. of aerosolized Mtb. After ten days to allow the acute infection to establish, chemotherapy was initiated. Mtb bacterial load was determined at the indicated time points. Data represent 3 mice at D0 (lung only) and 6 mice at D10 and D24.

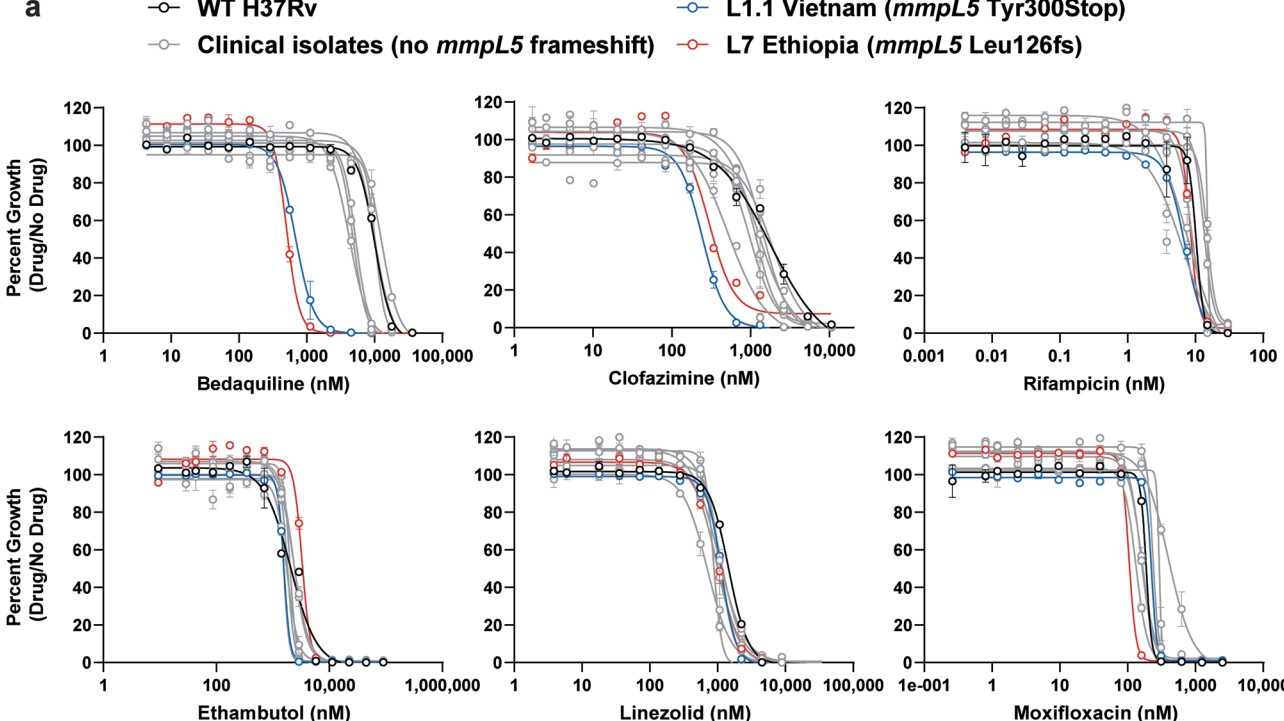

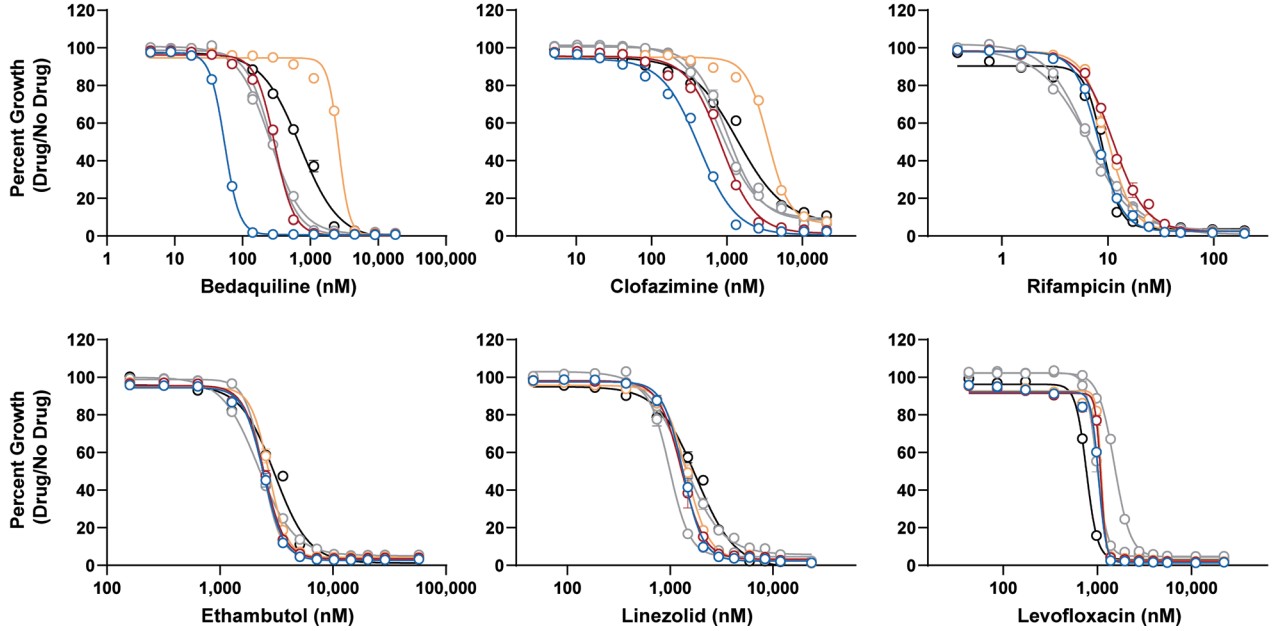

**Extended Data Fig. 10 | Some Mtb clinical strains have loss-of-function *mmpL5* mutations that render them hypersensitive to bedaquiline and clofazimine.** Dose-response curves (mean ± SEM, n = 3 biological replicates) for a reference set of Mtb clinical strains[66]. **b**, Dose-response curves (mean ± SEM, n = 3 biological replicates) for the listed Mtb strains. The L1.1 strain (*mmpL5* Tyr300Stop) in panel **a** was transformed with an integrating plasmid expressing either an empty vector (EV) or the *mmpS5* + *mmpL5* operon driven by the endogenous (pNative) or *hsp60* promoter.

# Reporting Summary

## Statistics

For all statistical analyses, confirm that the following items are present in the figure legend, table legend, main text, or Methods section.

| n/a | Confirmed | |
|---|---|---|
| ☐ | ☒ | The exact sample size (*n*) for each experimental group/condition, given as a discrete number and unit of measurement |
| ☐ | ☒ | A statement on whether measurements were taken from distinct samples or whether the same sample was measured repeatedly |
| ☐ | ☒ | The statistical test(s) used AND whether they are one- or two-sided<br>*Only common tests should be described solely by name; describe more complex techniques in the Methods section.* |
| ☒ | ☐ | A description of all covariates tested |
| ☒ | ☐ | A description of any assumptions or corrections, such as tests of normality and adjustment for multiple comparisons |
| ☐ | ☒ | A full description of the statistical parameters including central tendency (e.g. means) or other basic estimates (e.g. regression coefficient) AND variation (e.g. standard deviation) or associated estimates of uncertainty (e.g. confidence intervals) |
| ☐ | ☒ | For null hypothesis testing, the test statistic (e.g. *F*, *t*, *r*) with confidence intervals, effect sizes, degrees of freedom and *P* value noted<br>*Give P values as exact values whenever suitable.* |
| ☐ | ☒ | For Bayesian analysis, information on the choice of priors and Markov chain Monte Carlo settings |
| ☐ | ☒ | For hierarchical and complex designs, identification of the appropriate level for tests and full reporting of outcomes |
| ☐ | ☒ | Estimates of effect sizes (e.g. Cohen's *d*, Pearson's *r*), indicating how they were calculated |

*Our web collection on statistics for biologists contains articles on many of the points above.*

## Software and code

Policy information about availability of computer code

| Data collection | Data collection was done with custom scripts dependent on third-party tools |
|---|---|
| Data analysis | - GraphPad Prism (version 9.1.2)<br>- Microsoft Excel(365)<br>- MAGeCK analysis method (version 0.5.9.2) in python (version 2.7.16)<br>- subread-align (version 1.6.0)<br>- Sci Py package (version 1.2.2)<br>- FastTree (version 2.1.11 SSE3)<br>- Mykrobe(Version 0.9.0)<br>- GenomicAlignments (version 1.22.1)<br>- DESeq2 (version 1.30.1)<br>- Snippy (version 3.2-dev)<br>- QualiMap with the default parameters (version 2.2.2-dev)<br>- SpoTyping (version 2.1)<br>- iTol (https://itol.embl.de/)<br>- Rsamtools (Version 2.2.3)<br>- GenomicAlignments (Version 1.22.1)<br>- GenomicFeatures (Version 1.38.2)<br>- seqinr (Version 3.6-1)<br>- ggplot2 (Version 3.3.3)<br>- tidyverse (Version 1.3.1)<br>- genefilter (Version 1.68.0)<br>- Rsubread (Version 2.0.1) |

- Biostrings (Version 2.54.0)
- SummarizedExperiment (Version 1.16.1)
- BiocParallel (Version 1.20.1)
- pandas (Version 1.4.1)
- ast (Version 3.10.4)
- readr (Version 1.4.0)
- reshape2 (Version 1.4.4)
- gplots (Version 3.1.1)
- dendextend (Version 1.14.0)
- RColorBrewer (Version 1.1-2)
- openpyxl (Version 3.0.5)
- circlize (Version 0.4.12)

For manuscripts utilizing custom algorithms or software that are central to the research but not yet described in published literature, software must be made available to editors and reviewers. We strongly encourage code deposition in a community repository (e.g. GitHub). See the Nature Portfolio guidelines for submitting code & software for further information.

# Data

Policy information about availability of data

All manuscripts must include a data availability statement. This statement should provide the following information, where applicable:

- Accession codes, unique identifiers, or web links for publicly available datasets
- A description of any restrictions on data availability
- For clinical datasets or third party data, please ensure that the statement adheres to our policy

Raw sequencing data has been deposited to the Short Read Archive (SRA) under project number PRJNA738381. All screen results are available in Supplemental Data 1 and at pebble.rockefeller.edu. The custom code is available at https://github.com/rock-lab/CGI_nature_micro_2022.
H37Rv reference genome (NC_018143) was applied for alignment and SNPs calling. The accession numbers for previously sequenced strains used in Fig 5f, 6f, 8b are available in Source Data 4.

# Field-specific reporting

Please select the one below that is the best fit for your research. If you are not sure, read the appropriate sections before making your selection.

☒ Life sciences          ☐ Behavioural & social sciences          ☐ Ecological, evolutionary & environmental sciences

For a reference copy of the document with all sections, see nature.com/documents/nr-reporting-summary-flat.pdf

# Life sciences study design

All studies must disclose on these points even when the disclosure is negative.

| Sample size | The growth and drug dose-responsive curves presented in the manuscript consist of at least three technical replicates and two independent experiments. We used three biologically independent replicates for in vitro killing assays (Fig 2c, Extended Data Figure 4g, Extended Data Figure 5b). Mouse experiments included at least 5 mice per group which was sufficient to identify differences in bacterial burden between groups in previously published experiments. All graphs show mice as individual data points. |
|---|---|
| Data exclusions | Extended Data Figure 1c and 1h: Due to sample loss, in EDF 1c the EMB "Med" day 1 sequencing results reflect two replicates, one of which was sequenced in technical duplicate (to have 3 replicates total). In EDF 1h, STR "Med" samples harvested for deep sequencing only reflect the day 1 and day 5 timepoints because the 10 day samples was lost during sample preparation and DNA extraction. |
| Replication | We have indicated the number of times experiment was independently performed as described in the figure legends and their corresponding methods. |
| Randomization | There was no randomization in this study because there were no features for which randomization was deemed necessary. Mice used in this study are female BALB/c mice (6-8 weeks of age), which are inbred and should be genetically identical to each other. |
| Blinding | Blinding was not performed for any of the in vitro or in vivo experimentation. The measurements of optical density, CFU, RNA expression do not require researcher-based judgments and therefore we deemed blinding not necessary |

# Reporting for specific materials, systems and methods

We require information from authors about some types of materials, experimental systems and methods used in many studies. Here, indicate whether each material, system or method listed is relevant to your study. If you are not sure if a list item applies to your research, read the appropriate section before selecting a response.

## Materials & experimental systems

| n/a | Involved in the study |
|-----|----------------------|
| ☒ | Antibodies |
| ☒ | Eukaryotic cell lines |
| ☒ | Palaeontology and archaeology |
| ☐ | ☒ Animals and other organisms |
| ☒ | Human research participants |
| ☒ | Clinical data |
| ☒ | Dual use research of concern |

## Methods

| n/a | Involved in the study |
|-----|----------------------|
| ☒ | ChIP-seq |
| ☒ | Flow cytometry |
| ☒ | MRI-based neuroimaging |

# Animals and other organisms

Policy information about studies involving animals; ARRIVE guidelines recommended for reporting animal research

| | |
|---|---|
| Laboratory animals | Bone marrow-derived macrophages (BMDMs) were differentiated from wild-type, female C57BL/6NTAC mice (Taconic Farms, 6-8 weeks of age). 7-8 weeks-old Female BALB/c mice (Charles Rivers Laboratory) were infected with approximately 100-200 CFU of Mtb using a whole-body inhalation exposure system (Glas-Col) at the Biosafety Level-3(BSL-3) Laboratory. Mice were kept at 12h light/12h dark cycle at an ambient temperature of 72 degrees +/-2 and a relative humidity of 40-60% |
| Wild animals | The study did not involve wild animals. |
| Field-collected samples | The study did not involve samples collected from the field. |
| Ethics oversight | All animal work was performed in accordance with the Guide for the Care and Use of Laboratory Animals of the National Institutes of Health, with approval from the Institutional Animal Care and Use Committee of Rockefeller University and Committee of the Center for Discovery and Innovation. |

Note that full information on the approval of the study protocol must also be provided in the manuscript.

