## [Peer Review File · Nature Microbiology]

Peer Review Information

Journal: Nature Microbiology

Manuscript Title: CRISPRi chemical genetics and comparative genomics identify genes mediating drug potency in Mycobacterium tuberculosis

Corresponding author name(s): Jeremy Rock

Reviewer Comments & Decisions:

Decision Letter, initial version:

Dear Jeremy,

Thank you for your patience while your manuscript "A chemical-genetic map of the pathways controlling drug potency in *Mycobacterium tuberculosis*" was under peer-review at Nature Microbiology. It has now been seen by 3 referees, whose expertise and comments you will find at the of this email. You will see from their comments below that while they find your work of interest, some important points are raised. We are very interested in the possibility of publishing your study in Nature Microbiology, but would like to consider your response to these concerns in the form of a revised manuscript before we make a final decision on publication.

In particular, you will see that referee #3 has some concerns regarding the way their previous comments were addressed. The referee suggests to provide more thorough explanations or tone down some claims. The rest referees' reports are clear and the remaining issues should be straightforward to address.

If you have not done so already please begin to revise your manuscript so that it conforms to our Article format instructions at <http://www.nature.com/nmicrobiol/info/final-submission/>

The usual length limit for a Nature Microbiology Article is six display items (figures or tables) and 3,000 words. We have some flexibility, and can allow a revised manuscript at 3,500 words, but please consider this a firm upper limit. There is a trade-off of ~250 words per display item, so if you need more space, you could move a Figure or Table to Supplementary Information.

Some reduction could be achieved by focusing any introductory material and moving it to the start of your opening 'bold' paragraph, whose function is to outline the background to your work, describe in a sentence your new observations, and explain your main conclusions. The discussion should also be limited. Methods should be described in a separate section following the discussion, we do not place a word limit on Methods.

Nature Microbiology titles should give a sense of the main new findings of a manuscript, and should not contain punctuation. Please keep in mind that we strongly discourage active verbs in titles, and that they should ideally fit within 90 characters each (including spaces).

We strongly support public availability of data. Please place the data used in your paper into a public data repository, if one exists, or alternatively, present the data as Source Data or Supplementary Information. If data can only be shared

2nature portfolio

on request, please explain why in your Data Availability Statement, and also in the correspondence with your editor. For some data types, deposition in a public repository is mandatory - more information on our data deposition policies and available repositories can be found at <https://www.nature.com/nature-research/editorial-policies/reporting-standards#availability-of-data>.

Please include a data availability statement as a separate section after Methods but before references, under the heading "Data Availability". This section should inform readers about the availability of the data used to support the conclusions of your study. This information includes accession codes to public repositories (data banks for protein, DNA or RNA sequences, microarray, proteomics data etc...), references to source data published alongside the paper, unique identifiers such as URLs to data repository entries, or data set DOIs, and any other statement about data availability. At a minimum, you should include the following statement: "The data that support the findings of this study are available from the corresponding author upon request", mentioning any restrictions on availability. If DOIs are provided, we also strongly encourage including these in the Reference list (authors, title, publisher (repository name), identifier, year). For more guidance on how to write this section please see: <http://www.nature.com/authors/policies/data/data-availability-statements-data-citations.pdf>

To improve the accessibility of your paper to readers from other research areas, please pay particular attention to the wording of the paper's opening bold paragraph, which serves both as an introduction and as a brief, non-technical summary in about 150 words. If, however, you require one or two extra sentences to explain your work clearly, please include them even if the paragraph is over-length as a result. The opening paragraph should not contain references. Because scientists from other sub-disciplines will be interested in your results and their implications, it is important to explain essential but specialised terms concisely. We suggest you show your summary paragraph to colleagues in other fields to uncover any problematic concepts.

If your paper is accepted for publication, we will edit your display items electronically so they conform to our house style and will reproduce clearly in print. If necessary, we will re-size figures to fit single or double column width. If your figures contain several parts, the parts should form a neat rectangle when assembled. Choosing the right electronic format at this stage will speed up the processing of your paper and give the best possible results in print. We would like the figures to be supplied as vector files - EPS, PDF, AI or postscript (PS) file formats (not raster or bitmap files), preferably generated with vector-graphics software (Adobe Illustrator for example). Please try to ensure that all figures are non-flattened and fully editable. All images should be at least 300 dpi resolution (when figures are scaled to approximately the size that they are to be printed at) and in RGB colour format. Please do not submit Jpeg or flattened TIFF files. Please see also 'Guidelines for Electronic Submission of Figures' at the end of this letter for further detail.

Figure legends must provide a brief description of the figure and the symbols used, within 350 words, including definitions of any error bars employed in the figures.

nature portfolio

Please include a statement before the acknowledgements naming the author to whom correspondence and requests for materials should be addressed.

Finally, we require authors to include a statement of their individual contributions to the paper -- such as experimental work, project planning, data analysis, etc. -- immediately after the acknowledgements. The statement should be short, and refer to authors by their initials. For details please see the Authorship section of our joint Editorial policies at http://www.nature.com/authors/editorial_policies/authorship.html

- * include a point-by-point response to any editorial suggestions and to our referees. Please include your response to the editorial suggestions in your cover letter, and please upload your response to the referees as a separate document.

- * ensure it complies with our format requirements for Letters as set out in our guide to authors at www.nature.com/nmicrobiol/info/gta/

- * state in a cover note the length of the text, methods and legends; the number of references; number and estimated final size of figures and tables

- * resubmit electronically if possible using the link below to access your home page:

{redacted}

*This url links to your confidential homepage and associated information about manuscripts you may have submitted or be reviewing for us. If you wish to forward this e-mail to co-authors, please delete this link to your homepage first.

Please ensure that all correspondence is marked with your Nature Microbiology reference number in the subject line.

Nature Microbiology is committed to improving transparency in authorship. As part of our efforts in this direction, we are now requesting that all authors identified as 'corresponding author' on published papers create and link their Open Researcher and Contributor Identifier (ORCID) with their account on the Manuscript Tracking System (MTS), prior to acceptance. This applies to primary research papers only. ORCID helps the scientific community achieve unambiguous attribution of all scholarly contributions. You can create and link your ORCID from the home page of the MTS by clicking on 'Modify my Springer Nature account'. For more information please visit www.springernature.com/orcid

We hope to receive your revised paper within three weeks. If you cannot send it within this time, please let us know.

Yours sincerely,

{redacted}

Reviewer Expertise:

Referee #1: tuberculosis

Referee #2: Mtb physiology, metabolism, antimicrobials

Referee #3: CRISPR-based screens

Reviewers Comments:

Reviewer #1 (Remarks to the Author):

The authors have addressed all of my concerns with the previous version.

Reviewer #2 (Remarks to the Author):

The manuscript entitled "A chemical-genetic map of the pathways controlling drug potency in Mycobacterium tuberculosis" by Li and colleagues describes a massive, parallel chemical-genetic study aimed at mapping determinants of antibiotic potency in Mycobacterium tuberculosis. The results presented not only advance our understanding of fundamental aspects of M. tuberculosis antibiotic research, but also deliver on a potential treatment alternative to tuberculosis caused by specific strains. The massive amount of data within this manuscript will also likely trigger in depth analysis by a number of other groups, working on their favourite antibiotics. The work is at the forefront of molecular biology, employing the latest advances in CRISPRi regulation at genome scale.

The authors start by demonstrating the way their chemical-genetic profiling works (Figure 1), followed by a deeper analysis of five distinct sets of results: Figure 2 depicts the analysis of MtrAB-LpqB and envelope integrity; Figure 3 covers distinct pathways interfering with ribosome inhibition (a concept that is not novel per se, but the results in M. tuberculosis are); Figure 4 illustrates the effect of BacA on aminoglycosides and capreomycin; Figure 5 covers the interesting albeit low-level results on ettA and whiB7 stress response regulator (not new, as whiB7 has been implicated in antibiotic resistance in mycobacterial species, including M. tuberculosis); and Figure 6, describing a completely novel loss-of-function polymorphism (Gly64delG) on the Indo-Oceanic lineage, making these strains hypersusceptible to macrolides. This final section has of course direct, real life applications, on the treatment of strains belonging to sublineage 1, which are endemic in places such as Thailand, Philippines, and Indonesia.

Due to the elegant, state-of-the-art and comprehensive work described in this manuscript, it will likely become a reference study in chemical-genetic interaction in bacteria. And hence, I strongly support its publication in Nature Microbiology after minor revisions.

My only recommendation is to swap the word "controlling" and "control", used in the title and throughout the manuscript. What the authors are probing are alterations or interferences, sometimes indirect, due to CRISPR

interference of target genes. This is distinct from a gene product controlling a biological response. In some cases, the effects measured are as low as 2.3-fold change in MIC, that is hardly “control”.

Reviewer #3 (Remarks to the Author):

I don't think the authors have addressed my comments thoroughly. For example, how do you truncate the length of sgRNA and know the quantitative knockdown? If not, the authors should tone down their claim of 'predictability' of the approach. Also, assuming trans-complementation in Mtb is similar to *S. pneumoniae* is not totally accurate and lacks evidence. But they did provide reasonable explanations to questions related to the polarity effects and novelty of the approach, by referring to their published work, PMID: 34297925. This work seems more like an expansion of their published work PMID: 34297925. The overall scope of the work should be a fit to Nature Microbiology,

Author Rebuttal to Initial comments

Reviewer #1 (Remarks to the Author):

The authors have addressed all of my concerns with the previous version.

We thank the reviewer for their prior suggestions and their approval of the revised manuscript.

Reviewer #2 (Remarks to the Author):

The manuscript entitled “A chemical-genetic map of the pathways controlling drug potency in *Mycobacterium tuberculosis*” by Li and colleagues describes a massive, parallel chemical-genetic study aimed at mapping determinants of antibiotic potency in *Mycobacterium tuberculosis*. The results presented not only advance our understanding of fundamental aspects of *M. tuberculosis* antibiotic research, but also deliver on a potential treatment alternative to tuberculosis caused by specific strains. The massive amount of data within this manuscript will also likely trigger in depth analysis by a number of other groups, working on their favourite antibiotics. The work is at the forefront of molecular biology, employing the latest advances in CRISPRi regulation at genome scale.

The authors start by demonstrating the way their chemical-genetic profiling works (Figure 1), followed by a

deeper analysis of five distinct sets of results: Figure 2 depicts the analysis of MtrAB-LpqB and envelope integrity; Figure 3 covers distinct pathways interfering with ribosome inhibition (a concept that is not novel per se, but the results in *M. tuberculosis* are); Figure 4 illustrates the effect of BacA on aminoglycosides and capreomycin; Figure 5 covers the interesting albeit low-level results on *ettA* and *whiB7* stress response regulator (not new, as *whiB7* has been implicate in antibiotic resistance in mycobacterial species, including *M. tuberculosis*); and Figure 6, describing a completely novel loss-of-function polymorphism (Gly64delG) on the Indo-Oceanic lineage, making these strains hypersusceptible to macrolides. This final section has of course direct, real life applications, on the treatment of strains belonging to sublineage 1, which are endemic in places such as

Thailand, Philippines, and Indonesia.

Due to the elegant, state-of-the-art and comprehensive work described in this manuscript, it will likely become a reference study in chemical-genetic interaction in bacteria. And hence, I strongly support its publication in Nature Microbiology after minor revisions.

We thank the reviewer for their prior suggestions and their approval of the revised manuscript.

My only recommendation is to swap the word “controlling” and “control”, used in the title and throughout the manuscript. What the authors are probing are alterations or interferences, sometimes indirect, due to CRISPR interference of target genes. This is distinct from a gene product controlling a biological response. In some cases, the effects measured are as low as 2.3-fold change in MIC, that is hardly “control”.

The reviewer’s point is well taken. We removed all uses of the words “controlling” and “control” and replaced these with “influencing” and “influence.”

Reviewer #3 (Remarks to the Author):

I don’t think the authors have addressed my comments thoroughly. For example, how do you truncate the

length of sgRNA and know the quantitative knockdown? If not, the authors should tone down their claim of 'predictability' of the approach.

In our previous response to reviewers document we stated that the titration of gene expression in our system is described in PMID: 34297925. To briefly address this reviewer's comments here, we highlight below data presented in PMID: 34297925 that describes the predictable tunability of our CRISPRi system. But we encourage the reviewer to read PMID: 34297925 for a complete description of our approach, how we tune the magnitude of target knockdown, and data to support our claim that tunable knockdown is predictable.

We use the same Sth1dCas9 CRISPRi genome-scale library in PMID: 34297925 and in this chemical-genetics manuscript. Knockdown tuning with our CRISPRi system was achieved in two ways. First, we used the ability of Sth1dCas9 to recognize non-canonical protospacer adjacent motifs (PAMs) that lead to a gradient of target knockdown (PMID: 28165460, 34297925). Second, we varied the length of the sgRNA targeting sequence to modulate the extent of complementarity between the sgRNA and DNA target, which has been shown to influence target knockdown efficiency (PMID: 33545038, 34297925). In PMID: 34297925, we used this CRISPRi library to perform a competitive growth experiment over ~30 generations and quantified individual sgRNA abundance by deep sequencing. We then applied a linear model to determine which sgRNA features are most important in predicting the rate of sgRNA depletion during the competitive growth experiment (a proxy for sgRNA "strength"). We found that the targeted PAM, sgRNA targeting sequence length, and GC content were the dominant predictors of sgRNA strength (**Panels A,B**).

(A) Equations and parameters of the two-line model fit to quantify sgRNA depletion over time. Model fits and R^2 values for three different sgRNAs targeting *mmpL3* (*rv0206c*) are shown.

(B) Analysis of the relationship between sgRNA features and sgRNA rate of depletion (β_e) using a linear model identified three features (targeted PAM, sgRNA targeting sequence length, and GC content) that contribute to sgRNA efficacy. Bars show regression coefficients (mean \pm SEM) for each feature colored by feature type. All features were represented by more than 500 sgRNAs except for the 20-30% GC ($n=18$) and 90-100% GC ($n=458$) bins.

To validate these sgRNA strength predictions, we designed sgRNAs of varying predicted strengths and measured how strongly they reduced expression of a luminescent reporter gene (*Renilla luciferase*) in *M. smegmatis*. We found a strong correlation ($R^2=0.74$) between predicted sgRNA strength and *Renilla* knockdown (**Panel C**). That the linear model was trained on fitness phenotypes in *Mtb* and accurately predicted *Renilla* knockdown values in *M. smegmatis* supports the hypothesis that CRISPRi knockdown efficacy is, at least in part, determined by biophysical parameters of the dCas9-sgRNA-DNA interaction. Having validated the linear model predictions, we next normalized sgRNA strength predictions to span values from 0 (weakest; blue) to 1 (strongest; red). The growth effects for sgRNAs of varying predicted strengths targeting an essential (*mmpL3*) and non-essential (*clgR*) gene (**Panel D**) generally matched the expected phenotypes, providing visual confirmation of the sgRNA strength predictions and further demonstrating the broad tunability of target gene knockdown with this CRISPRi system.

(C) Comparison of measured versus predicted CRISPRi activity against a *Renilla luciferase* target in *M. smegmatis*. The linear model was used to generate sgRNA strength predictions for 29 sgRNAs targeting the *Renilla* ORF (predicted strength range: 0.018 - 0.973); color-coded from blue (strength=0) to red (strength=1). The green dot shows the RLU fold change for a control non-targeting sgRNA. RLU=relative light units.

(D) Line plot showing the behavior of all sgRNAs targeting the essential gene *mmpL3* (*rv0206c*) and non-essential gene *clgR* (*rv2745c*). sgRNAs are color-coded by predicted strengths (0=blue - 1=red). Circles represent our sequencing limit of detection. Triangles represent the point of observation of rare CRISPRi-resistant subpopulations, beyond which sgRNA L2FC values are not plotted.

Please see PMID: 34297925 for much more detail describing the CRISPRi system, sgRNA strength predictions, and experimental validation of predictable tunability. We also highlight the work from others using a mismatch CRISPRi approach— distinct from what we describe here— for predictable, tunable knockdown (PMID: 33221881, 34027480, 33080209).

Also, assuming trans-complementation in Mtb is similar to *S. pneumoniae* is not totally accurate and lacks evidence.

We agree that the extent of trans-complementation may be different for Mtb and *S. pneumoniae*. But *S. pneumoniae* is the only organism for which the question “How prevalent is trans-complementation?” has been systematically addressed— and it is rare, as reported by PMID: 31844066.

To address this reviewers concern, we stress in our revised Discussion section that: “Caveats include that... all pooled screens may miss effects where the phenotype can be complemented in trans (e.g. cross-feeding).”

But they did provide reasonable explanations to questions related to the polarity effects and novelty of the approach, by referring to their published work, PMID: 34297925. This work seems more like an expansion of their published work PMID: 34297925. The overall scope of the work should be a fit to Nature Microbiology,

We thank the reviewer for their support of our work.

Decision Letter, first revision:

Dear Jeremy,

Thank you for submitting your revised manuscript "A chemical-genetic map of the pathways influencing drug potency in *Mycobacterium tuberculosis*" (NMICROBIOL-22010206A). After discussing your rebuttal letter with the editorial team, we find that the paper has improved in revision, and that you properly addressed the points raised by the referees. Therefore we'll be happy in principle to publish it in Nature Microbiology, pending minor revisions to satisfy the referees' final requests and to comply with our editorial and formatting guidelines.

Thank you again for your interest in Nature Microbiology Please do not hesitate to contact me if you have any questions.

2Sincerely,

{redacted}

Decision Letter, final checks:

Dear Jeremy,

Thank you for your patience as we've prepared the guidelines for final submission of your Nature Microbiology manuscript, "A chemical-genetic map of the pathways influencing drug potency in *Mycobacterium tuberculosis*" (NMICROBIOL-22010206A). Please carefully follow the step-by-step instructions provided in the attached file, and add a response in each row of the table to indicate the changes that you have made. Please also check and comment on any additional marked-up edits we have proposed within the text. Ensuring that each point is addressed will help to ensure that your revised manuscript can be swiftly handed over to our production team.

In recognition of the time and expertise our reviewers provide to Nature Microbiology's editorial process, we would like to formally acknowledge their contribution to the external peer review of your manuscript entitled "A chemical-genetic map of the pathways influencing drug potency in *Mycobacterium tuberculosis*". For those reviewers who give their assent, we will be publishing their names alongside the published article.

Nature Microbiology offers a Transparent Peer Review option for new original research manuscripts submitted after December 1st, 2019. As part of this initiative, we encourage our authors to support increased transparency into the peer review process by agreeing to have the reviewer comments, author rebuttal letters, and editorial decision letters published as a Supplementary item. When you submit your final files please clearly state in your cover letter whether or not you would like to participate in this initiative. Please note that failure to state your preference will result in delays in accepting your manuscript for publication.

3Cover suggestions

As you prepare your final files we encourage you to consider whether you have any images or illustrations that may be appropriate for use on the cover of Nature Microbiology.

Nature Microbiology has now transitioned to a unified Rights Collection system which will allow our Author Services team to quickly and easily collect the rights and permissions required to publish your work. Approximately 10 days after your paper is formally accepted, you will receive an email in providing you with a link to complete the grant of rights. If your paper is eligible for Open Access, our Author Services team will also be in touch regarding any additional information that may be required to arrange payment for your article.

Please note that *Nature Microbiology* is a Transformative Journal (TJ). Authors may publish their research with us through the traditional subscription access route or make their paper immediately open access through payment of an article-processing charge (APC). Authors will not be required to make a final decision about access to their article until it has been accepted. [Find out more about Transformative Journals](https://www.springernature.com/gp/open-research/transformative-journals)

Authors may need to take specific actions to achieve [compliance with funder and institutional open access mandates](https://www.springernature.com/gp/open-research/funding/policy-compliance-faqs). If your research is supported by a funder that requires immediate open access (e.g. according to [Plan S principles](https://www.springernature.com/gp/open-research/plan-s-compliance)) then you should select the gold OA route, and we will direct you to the compliant route where possible. For authors selecting the subscription publication route, the journal's standard licensing terms will need to be accepted, including [self-archiving policies](https://www.springernature.com/gp/open-research/policies/journal-policies). Those licensing terms will supersede

4any other terms that the author or any third party may assert apply to any version of the manuscript.

For information regarding our different publishing models please see our <https://www.springernature.com/gp/open-research/transformative-journals> Transformative Journals page. If you have any questions about costs, Open Access requirements, or our legal forms, please contact ASJournals@springernature.com.

Please use the following link for uploading these materials:
{redacted}

If you have any further questions, please feel free to contact me.
{redacted}

Final Decision Letter:

Dear Jeremy,

I am very pleased to accept your Article "CRISPRi chemical genetics and comparative genomics identify genes mediating drug potency in *Mycobacterium tuberculosis*" for publication in Nature Microbiology. Thank you for having chosen to submit your work to us and many congratulations.

Acceptance of your manuscript is conditional on all authors' agreement with our publication policies (see <https://www.nature.com/nmicrobiol/editorial-policies>). In particular your manuscript must not be published elsewhere and there must be no announcement of the work to any media outlet until the

5publication date (the day on which it is uploaded onto our website).

Please note that *Nature Microbiology* is a Transformative Journal (TJ). Authors may publish their research with us through the traditional subscription access route or make their paper immediately open access through payment of an article-processing charge (APC). Authors will not be required to make a final decision about access to their article until it has been accepted. [Find out more about Transformative Journals](https://www.springernature.com/gp/open-research/transformative-journals)

Authors may need to take specific actions to achieve [compliance with funder and institutional open access mandates](https://www.springernature.com/gp/open-research/funding/policy-compliance-faq). If your research is supported by a funder that requires immediate open access (e.g. according to [Plan S principles](https://www.springernature.com/gp/open-research/plan-s-compliance)) then you should select the gold OA route, and we will direct you to the compliant route where possible. For authors selecting the subscription publication route, the journal's standard licensing terms will need to be accepted, including [self-archiving policies](https://www.nature.com/nature-portfolio/editorial-policies/self-archiving-and-license-to-publish). Those licensing terms will supersede any other terms that the author or any third party may assert apply to any version of the manuscript.

To assist our authors in disseminating their research to the broader community, our SharedIt initiative

6nature portfolio

provides you with a unique shareable link that will allow anyone (with or without a subscription) to read the published article. Recipients of the link with a subscription will also be able to download and print the PDF.
